# Teacher Ascent: Robust and Efficient Machine Unlearning via Knowledge Distillation and Continual Learning

## Abstract

Removing specific knowledge from a trained machine learning model is an open problem of increasing importance. Growing dataset sizes increase the likelihood of introducing biased, inaccurate, or private data. Moreover, increasing the number of parameters makes retraining models more costly. While powerful Machine Unlearning methods have emerged as effective alternatives to retraining, their practical application is often hindered by narrow functional ranges for hyperparameters, which typically require access to a retrained model for effective tuning. Established unlearning methods like SCRUB+R and SSD require precise specification of their hyperparameters to achieve unlearning whilst preventing catastrophic forgetting. We address this challenge by proposing Teacher Ascent (TA), a novel unlearning method that is based on knowledge distillation and continual learning. Inspired by Elastic Weight Consolidation (EWC), TA forgets target data while protecting parameters essential for generalization by using the Fisher Information Matrix. We conduct experiments on MNIST, CIFAR, and Pins Face Recognition across various unlearning scenarios: forgetting entire classes, subclasses, and mislabeled samples. Our results demonstrate that Teacher Ascent both mimics the functional behavior of a retrained model across unlearning tasks while being 6-19 times more efficient than retraining. More importantly, TA mitigates catastrophic forgetting and demonstrates robustness across a wide range of hyperparameters. By overcoming the critical stability and tuning challenges of previous approaches, Teacher Ascent represents a significant step towards making machine unlearning a viable and practical tool for real-world applications.

## 1 Introduction

As machine learning models grow in scale and become more integrated in society, their capacity to internalize and reproduce data presents significant legal and ethical challenges. Large models have been found to generate outputs containing proprietary or restricted content, and they often "memorize" specific training data points (Carlini et al., 2019; Zhou et al., 2024). This behavior has led to high-profile copyright infringement lawsuits such as those initiated by Getty Images (Brittain & Brittain, 2023) and The New York Times, which argue that generative AI models illegally store and regurgitate protected material (Cooper & Grimmelmann, 2024). The regulatory pressure has been intensified globally with privacy frameworks like the European Union's General Data Protection Regulation (European Parliament & Council of the European Union, 2016) with its "right to be forgotten", California's Consumer Privacy Act (CCPA) (Chau, 2018), and Brazil's Data Protection Law (LGPD) (Brazilian National Congress, 2018). Concurrently, broader frameworks like the EU's Artificial Intelligence Act aim to mitigate systemic risks by requiring model providers to prevent or minimize harmful or undesirable behavior (European Parliament & Council of the European Union, 2024). Together, these legal and safety requirements create a need for methods that can modify already trained and deployed models without the prohibitive cost of a complete retraining.

One emerging field that addresses this need is Machine Unlearning (MU) Bourtoule et al. (2021). Formally, we assume a model $\mathcal{M}_{\boldsymbol{\theta}} : \mathbb{R}^{d^{(0)}} \to \mathbb{R}^C$ with parameters $\boldsymbol{\theta}$ has been trained on a dataset $\mathcal{D} = \{(\boldsymbol{x}_i, y_i)\}_{i=1}^N$. Here, $d^{(0)}$ represents the input feature dimension and $C$ is the number of classes. The objective is to remove the influence of a *forget set*, $\mathcal{D}_f \subset \mathcal{D}$, while preserving performance

on the remaining dataset, called the *retain set*, $\mathcal{D}_r = \mathcal{D} \backslash \mathcal{D}_f$. The ultimate goal is to produce an unlearned model that is functionally equivalent to a model trained from scratch on the retain set $\mathcal{D}_r$.

The field of MU is broad, and many paradigms exist. While exact unlearning methods (Bourtoule et al., 2021; Yan et al., 2022) offer provable guarantees of data removal, they require accounting for unlearning during initial model training, limiting their use. This has motivated a shift towards approximate unlearning, which relaxes removal guarantees in favor of making unlearning applicable to a broader class of models. This paper focuses on a common practical scenario within approximate unlearning. We assume a "full-access" setting where both the retain and forget sets are available at unlearning time, as opposed to zero-shot (Chundawat et al., 2023) or zero-glance (Tarun et al., 2024; Zhou et al., 2025) approaches where data access is restricted at the time of unlearning (Nguyen et al., 2022). Our work targets sample-level unlearning, i.e., the removal of individual samples or batches of samples, which can be easily extended to entire classes. This scope allows us to develop a practical fine-tuning solution for modifying large, pre-existing models.

Several prior unlearning methods fall under this setting. Of particular interest is SCalable Remembering and Unlearning unBound + Rewind (SCRUB+R) (Kurmanji et al., 2023), a fine-tuning method that seeks to preserve model performance while forgetting select data. Another noteworthy method is Selective Synaptic Dampening (SSD) (Foster et al., 2024), which seeks to identify and intervene on parameters specialized to the forget set. While both have shown promising performance, a key limitation is hyperparameter sensitivity. Specifically, SCRUB+R converges towards catastrophic forgetting if run for too long, a result of maximizing an *unbounded* KL-divergence term. Although this can seem like an implementation detail, the authors stress that the maximization step should be performed for "a few epochs in practice" and incorporate a rewind procedure, also to mitigate catastrophic forgetting. Meanwhile, SSD is highly dependent on the predefined threshold at which parameters are intervened on. Setting the threshold too low results in model degradation, and too high results in performing no model update at all. Crucially, choosing these hyperparameters appropriately is dependent on the forget set.

In this paper, we propose Teacher Ascent (TA), a fine-tuning based unlearning method built on principles from knowledge distillation and continual learning. TA consistently tracks the functional behavior of a retrained model across several benchmarks and forget sets while remaining far more efficient than retraining. Furthermore, TA exhibits high robustness to the choice of its hyperparameters making it applicable to practical unlearning scenarios.

We achieve this, in part, by maximizing *bounded* KL divergence terms during removal of $\mathcal{D}_f$, thereby circumventing catastrophic forgetting. When forgetting, a regularization term, inspired by Elastic Weight Consolidation (EWC) (Kirkpatrick et al., 2017), protects parameters important for the retain set. To further protect knowledge about $\mathcal{D}_r$, an additional objective that encourages similar behavior to the original model on this dataset is optimized. Across the considered benchmarks, we find that catastrophic forgetting can be mitigated by sampling few minibatches from $\mathcal{D}_r$. This observation was key in making TA efficient compared to a retrained model.

### 1.1 Contributions

We list the three main contributions:

- We propose Teacher Ascent (TA), an efficient unlearning method that consistently tracks the behavior of a retrained model and exhibits robustness to the choice of its hyperparameters.
- We demonstrate that the established fine-tuning method, SCRUB+R converges to catastrophic forgetting, highlighting a critical reliability gap in existing approaches.
- We propose a more realistic evaluation protocol by searching for hyperparameters on a semantically related unlearning task. This is aimed at highlighting hyperparameter sensitivity, a key gap between current unlearning literature and practical forgetting requests.

### 1.2 Related Work

**Fine-tuning and Knowledge Distillation:** A promising paradigm in approximate unlearning involves fine-tuning a model to erase the influence of specific data. One branch of this research relies

on training auxiliary models. These approaches include training an "incompetent teacher" to guide the unlearning process (Chundawat S et al., 2023), subtracting the output logits from a model trained to perform well on $\mathcal{D}_f$ (Ji et al., 2024), or aligning knowledge gaps with models trained on external data (Wang et al., 2023). While often effective, this reliance on auxiliary models introduces significant overhead and complicates evaluation. Other methods modify the original model more directly. DELETE Zhou et al. (2025) decompose the loss into forget and retain terms, suppressing probability mass from forget set classes whilst preserving the relative probabilities among retained classes. Spartalis et al. (2025) propose LoTUS which smoothens the predicted probabilities up to an information theoretic bound to mimic the confidence of a retrained model on $\mathcal{D}_f$. (Tarun et al., 2024) use an impair and repair strategy, first degrading the model's performance on the forget set through techniques like targeted noise injection, and then recovering general performance by fine-tuning on the retain set. Similarly, Amnesiac Unlearning (Graves et al., 2021) reverses the learning process by subtracting stored parameter updates, but it is practicality limited by prohibitive storage costs and the necessity of its own repair phase. SCRUB+R (Kurmanji et al., 2023) follow a teacher-student framework and design an objective to make the model diverge on forget data while preserving knowledge about $\mathcal{D}_r$.

**Fisher Information:** Multiple existing methods use Fisher Information in an unlearning context. For a multivariate model that has converged to the optimal parameters, the Fisher Information Matrix (FIM) is defined as the covariance of the score function, i.e., the gradient of the log-likelihood. Diagonal FIM elements quantify how much information about the dataset is captured in each parameter, while the off-diagonal entries measure how strongly two parameters' effects on the likelihood are correlated. Hence, large off-diagonal values indicate that the parameters are not independently identifiable from the data. SSD (Foster et al., 2024) use the diagonal of the empirical FIM with respect to retain and forget data to quantify how much more information a parameter contains about $\mathcal{D}_f$ versus $\mathcal{D}_r$. If this exceeds a pre-defined threshold, that parameter is intervened on. Golatkar et al. (2020) propose Fisher Forgetting which perturbs parameters with Gaussian noise with a variance inversely proportional to how important the parameter is for retain data.

**Continual Learning:** Continual learning is a field concerned with learning a new task without catastrophically forgetting previously learned knowledge. EWC (Kirkpatrick et al., 2017) is a canonical approach which computes the diagonal of the empirical Fisher Information Matrix with respect to a previously learned dataset. When learning the new task, the distance between current and previous task parameters is minimized, weighted by the corresponding Fisher Information. In the context of unlearning, Zhang et al. (2023) build on EWC and fine-tune with Fisher penalties to selectively degrade the forget set performance while preserving retain set knowledge. Wang et al. (2024) use EWC while performing gradient ascent for a generated image to protect generalization. The resulting model is used downstream to assess which training images are forgotten, allowing one to quantify which images from the data distribution influenced the synthesized image.

## 2 BACKGROUND

### 2.1 SCRUB+R

SCRUB+R builds on a teacher-student framework where the original model, $\mathcal{M}_{\boldsymbol{\theta}_o}$, acts as the teacher and the unlearned model, $\mathcal{M}_{\boldsymbol{\theta}_u}$, is the student. The method works by maximizing the distance between student and teacher probabilities on $\mathcal{D}_f$ while staying close to the teacher on $\mathcal{D}_r$. To measure distances between probability distributions, temperature-scaled Kullback-Leibler divergence is used as presented in Hinton et al. (2014). Given unnormalized logits from the teacher model $\boldsymbol{p}$, and the student model $\boldsymbol{q}$ (where $\boldsymbol{p}, \boldsymbol{q} \in \mathbb{R}^C$), the first step uses the tempered softmax, where $\tau \in \mathbb{R}^+$:

$$\boldsymbol{p}_\tau = \text{softmax}\left(\frac{\boldsymbol{p}}{\tau}\right), \quad \boldsymbol{q}_\tau = \text{softmax}\left(\frac{\boldsymbol{q}}{\tau}\right) \tag{1}$$

The knowledge distillation loss is then defined as the KL-divergence, $D_{KL}$, between these softened distributions, scaled by $\tau^2$:

$$\mathcal{L}_{KD}(\boldsymbol{p}, \boldsymbol{q}, \tau) = \tau^2 \cdot D_{KL}(\boldsymbol{p}_\tau \| \boldsymbol{q}_\tau) \tag{2}$$

To induce forgetting, part of the SCRUB+R objective maximizes the distilled KL-divergence between teacher and student predictions on $\mathcal{D}_f$:

$$\mathcal{L}_f(\mathcal{M}_{\boldsymbol{\theta}_u}; \mathcal{M}_{\boldsymbol{\theta}_o}, \mathcal{D}_f) = -\frac{1}{|\mathcal{D}_f|} \sum_{\boldsymbol{x} \in \mathcal{D}_f} \mathcal{L}_{KD}(\mathcal{M}_{\boldsymbol{\theta}_o}(\boldsymbol{x}), \mathcal{M}_{\boldsymbol{\theta}_u}(\boldsymbol{x}), \tau_f)$$

where $\tau_f$ is a hyperparameter. Optimizing $\mathcal{L}_f$ in isolation leads to model degradation on $\mathcal{D}_r$. To this end, the authors propose a repair step where they minimize the cross-entropy along with $\mathcal{L}_{KD}$ between student and teacher predictions on $\mathcal{D}_r$. Formally, the repair loss becomes:

$$\mathcal{L}_{\text{repair}}(\mathcal{M}_{\boldsymbol{\theta}_u}; \mathcal{M}_{\boldsymbol{\theta}_o}, \mathcal{D}_r) = \frac{1}{|\mathcal{D}_r|} \sum_{(\boldsymbol{x},y) \in \mathcal{D}_r} \mathcal{L}_{CE}(\mathcal{M}_{\boldsymbol{\theta}_u}(\boldsymbol{x}), y) + \mathcal{L}_{KD}(\mathcal{M}_{\boldsymbol{\theta}_o}(\boldsymbol{x}), \mathcal{M}_{\boldsymbol{\theta}_u}(\boldsymbol{x}), \tau_r) \quad (3)$$

Where $\mathcal{L}_{CE}$ denotes the cross-entropy loss:

$$\mathcal{L}_{\text{CE}}(\boldsymbol{x}, y; \mathcal{M}_{\boldsymbol{\theta}}) = -\log\left(\text{softmax}(\mathcal{M}_{\boldsymbol{\theta}}(\boldsymbol{x}))\right)_y$$

Due to the conflicting nature of $\mathcal{L}_f$ and $\mathcal{L}_{repair}$, they are optimized in an alternating fashion similar to Goodfellow et al. (2020). Finally, to close any knowledge gaps between what a model trained on $\mathcal{D}_r$ could generalize to on $\mathcal{D}_f$, a sequence of steps where only $\mathcal{L}_{repair}$ is minimized are carried out.

While this procedure can mimic the behavior of a retrained model on some unlearning tasks, the authors observe that it can still be prone to "over-forgetting" e.g., suspiciously poor performance on the forget set. To mitigate this, they proposed an additional rewind step to restore a previous model state. Specifically, they sample a rewind set $\mathcal{D}_{rewind}$ from the holdout validation set that is of the same label distribution as $\mathcal{D}_f$. They then calculate the error of the model obtained after performing alternating optimization on $\mathcal{D}_{rewind}$ and store this as a reference point. The final model is chosen as the one whose error on the forget set is as close to the reference point as possible.

### 2.1.1 SSD

The SSD method seeks to identify parameters highly specialized to $\mathcal{D}_f$ and intervene on these. This is done post-hoc and hence no fine-tuning of the original model is performed. To quantify parameter importance with respect to a dataset, the diagonal of the empirical FIM (Schraudolph, 2002; Martens, 2020) is used. Formally, given a vector of model parameters $\boldsymbol{\theta}$ and dataset $D$, the diagonal of the empirical FIM is given as:

$$F(\boldsymbol{\theta}, D) = \frac{1}{|D|} \sum_{(\boldsymbol{x},y) \in D} \nabla_{\boldsymbol{\theta}} \log p(y|\boldsymbol{x}, \boldsymbol{\theta}) \odot \nabla_{\boldsymbol{\theta}} \log p(y|\boldsymbol{x}, \boldsymbol{\theta}) \quad (4)$$

Where $p(y|\boldsymbol{x}, \boldsymbol{\theta})$ is the model's predicted probability of class $y$ for input $\boldsymbol{x}$ and $\odot$ denotes the Hadamard product. To assess parameter importances, the authors compare entries in $\boldsymbol{f}^{(\mathcal{D}_f)} = F(\boldsymbol{\theta}_o, \mathcal{D}_f)$ and $\boldsymbol{f}^{(\mathcal{D}_r)} = F(\boldsymbol{\theta}_o, \mathcal{D}_r)$. Using these, a parameter, $\theta_j$ is intervened on according to the following rule:

$$\theta_j = \begin{cases} \beta \cdot \theta_j & f_j^{(\mathcal{D}_f)} > \alpha f_j^{(\mathcal{D}_r)} \\ \theta_j & \text{otherwise} \end{cases}$$

where $\alpha \in \mathbb{R}^+$ is a hyperparameter determining the threshold for intervention. Here, the dampening factor $\beta \in [0, 1]$ is calculated as:

$$\beta = \min\left(\frac{\lambda \cdot f_j^{(\mathcal{D}_r)}}{f_j^{(\mathcal{D}_f)}}, 1\right)$$

Here $\lambda \in \mathbb{R}^+$ is a hyperparameter controlling how strongly parameters should be dampened.

## 3 METHODS

Teacher Ascent follows a teacher-student paradigm similar to SCRUB+R. The goal is to encourage similar behavior to the original model on retain data while removing knowledge about the forget

set that a retrained model cannot generalize to. Like Kurmanji et al. (2023), we use distilled KL-divergence (Equation 2) but with the key difference that we bound the probabilities that serve as input to the KL-divergence to $\epsilon > 0$[1] as:

$$\tilde{q}_i = \max(\epsilon, q_i) \tag{5}$$

This step is crucial to mitigating catastrophic forgetting. To see why, we first consider the case of unbounded maximization of Equation 2. Given that the activation function of $\mathcal{M}$ has an unbounded codomain, the norm of $\boldsymbol{\theta}_u$ will diverge.

**Proposition 1** (Parameter norm diverges without bounding). *Maximizing $D_{KL}(\boldsymbol{p}_\tau \| \boldsymbol{q}_\tau)$ without bounding components of $\boldsymbol{q}_\tau$ admits no finite critical point for student logits. Consequently, gradient ascent causes the norm of the student parameters to diverge $\|\boldsymbol{\theta}_u\| \to \infty$.*

The derivations and proof of Proposition 1 are provided in Appendix A. Next, consider the case where student probabilities are bounded using Equation 5. Let $\mathcal{A} = \{k | q_k > \epsilon\}$ denote the set of active indices where student probabilities are not bounded. Let $P_\mathcal{A} = \sum_{k \in \mathcal{A}} p_k$ be the total probability mass of the teacher for active indices. Then in the unlearning limit, where the student has shifted probability mass from all classes where $p_k > 0$, the gradient norm will be zero. This property is formalized in Corollary 1 below:

**Corollary 1** (Gradient vanishing in the unlearning limit). *If the student suppresses all teacher-supported classes, i.e., $q_k \leq \epsilon$ for all $k$ where $p_k > 0$, then $P_\mathcal{A} \to 0$ and*

$$\lim_{P_\mathcal{A} \to 0} \left\| \frac{\partial D_{KL}(\boldsymbol{p}_\tau \| \tilde{\boldsymbol{q}}_\tau)}{\partial \boldsymbol{q}} \right\| = 0.$$

An immediate consequence of this is that the norm of $\boldsymbol{\theta}_u$ will remain finite. Further details can be found in Appendix A. The distilled KL-divergence with bounded probabilities is used to form the part of the objective in charge of confusing the unlearned model about the forget set. This term is formulated directly using the logit outputs from the teacher model, $\mathcal{M}_{\boldsymbol{\theta}_o}(\boldsymbol{x})$, and the student model, $\mathcal{M}_{\boldsymbol{\theta}_u}(\boldsymbol{x})$.

$$\mathcal{L}_{\text{unlearn}}(\mathcal{M}_{\boldsymbol{\theta}_u}; \mathcal{M}_{\boldsymbol{\theta}_o}, \mathcal{D}_f) = \frac{1}{|\mathcal{D}_f|} \sum_{\boldsymbol{x} \in \mathcal{D}_f} [\mathcal{L}_{KD}(\mathcal{M}_{\boldsymbol{\theta}_u}(\boldsymbol{x}), \mathbf{1}, \tau_e) - \mathcal{L}_{KD}(\mathcal{M}_{\boldsymbol{\theta}_o}(\boldsymbol{x}), \mathcal{M}_{\boldsymbol{\theta}_u}(\boldsymbol{x}), \tau_f)]$$

$$\tag{6}$$

The first term pushes the student's predictions towards a uniform distribution by using a target logit vector of all ones, $\mathbf{1}$, (representing maximum uncertainty). This corresponds to maximizing the Shannon entropy of the student's temperature-scaled probabilities on the forget set. The second term actively maximizes the divergence from the teacher's original predictions. $\tau_e, \tau_f \in \mathbb{R}^+$ are temperature hyperparameters.

While minimizing $\mathcal{L}_{\text{unlearn}}$ during the forgetting phase can lead to effective unlearning, we found this to be unstable without further safeguarding (see appendix C). To improve stability, we introduce a regularization term inspired by Elastic Weight Consolidation (EWC) (Kirkpatrick et al., 2017). EWC protects essential knowledge by penalizing large changes to model parameters that are critical for performance on the retain set i.e., it reduces the plasticity of parameters identified as crucial for performance on the retain set. This is achieved by minimizing a weighted distance between the original parameters $\boldsymbol{\theta}_o$ and the updated parameters $\boldsymbol{\theta}_u$. The weights are determined using the diagonal FIM (Kirkpatrick et al., 2017) given in Equation 4.

Early experiments showed that simply using the parameter importance derived from $\mathcal{D}_r$ was not an adequate regularizer. Instead, we propose a more discriminative approach that computes importance as a **ratio** of the diagonal FIM between $\mathcal{D}_r$ and $\mathcal{D}_f$. Informally, this ratio quantifies how much more information a parameter captures about retain data than forget data. Defining $\boldsymbol{f}^{(\mathcal{D}_r)} = F(\boldsymbol{\theta}_o, \mathcal{D}_r)$ and $\boldsymbol{f}^{(\mathcal{D}_f)} = F(\boldsymbol{\theta}_o, \mathcal{D}_f)$, the regularization term becomes:

$$\mathcal{L}_{\text{EWC}}(\mathcal{M}_{\boldsymbol{\theta}_u}; \mathcal{M}_{\boldsymbol{\theta}_o}, \boldsymbol{f}^{(\mathcal{D}_r)}, \boldsymbol{f}^{(\mathcal{D}_f)}) = \sum_j \frac{f_j^{(\mathcal{D}_r)}}{f_j^{(\mathcal{D}_f)}} (\theta_{u,j} - \theta_{o,j})^2 \tag{7}$$

Here, $f_j^{(\mathcal{D}_r)}$ and $f_j^{(\mathcal{D}_f)}$ are the $j$-th components of the FIM vectors $\boldsymbol{f}^{(\mathcal{D}_r)}$ and $\boldsymbol{f}^{(\mathcal{D}_f)}$, respectively

---

[1]We set $\epsilon = 10^{-8}$ in all experiments.

---

**Algorithm 1** Teacher Ascent Optimization procedure

---

1: **Input:** Original model $\mathcal{M}_{\boldsymbol{\theta}_o}$, forget set $\mathcal{D}_f$, retain set $\mathcal{D}_r$, batch size $b$, total rounds $R$, total forget rounds $R_f$, EWC strength $\lambda$, repair multiplier $k$, step size $\eta$.
2: **Initialize:** Unlearned model $\mathcal{M}_{\boldsymbol{\theta}_u} \leftarrow \mathcal{M}_{\boldsymbol{\theta}_o}$.
3: $\boldsymbol{f}^{(\mathcal{D}_r)} \leftarrow F(\boldsymbol{\theta}_o, \mathcal{D}_r)$
4: $\boldsymbol{f}^{(\mathcal{D}_f)} \leftarrow F(\boldsymbol{\theta}_o, \mathcal{D}_f)$
5: Define $n_f \leftarrow \lceil |\mathcal{D}_f|/b \rceil$  ▷ Number of forget steps per round
6: **for** $i$ from 1 to $R$ **do**
7:     **if** $i \leq R_f$ **then**
8:         **for** each minibatch $\mathcal{B}_f$ in $\mathcal{D}_f$ **do**  ▷ Sample all minibatches from forget set
9:             $\boldsymbol{\theta}_u \leftarrow \boldsymbol{\theta}_u - \eta \nabla_{\boldsymbol{\theta}_u} \mathcal{L}_{\text{forget}}(\mathcal{M}_{\boldsymbol{\theta}_u}; \mathcal{M}_{\boldsymbol{\theta}_o}, \mathcal{D}_f, \mathcal{D}_r)$
10:         **end for**
11:     **end if**
12:     **for** $j$ from 1 to $n_f \cdot k$ **do**
13:         Sample minibatch $\mathcal{B}_r$ from $\mathcal{D}_r$
14:         $\boldsymbol{\theta}_u \leftarrow \boldsymbol{\theta}_u - \eta \nabla_{\boldsymbol{\theta}_u} \mathcal{L}_{\text{repair}}(\mathcal{M}_{\boldsymbol{\theta}_u}; \mathcal{M}_{\boldsymbol{\theta}_o}, \mathcal{B}_r)$
15:     **end for**
16: **end for**
17: **return** $\boldsymbol{\theta}_u$

---

while $\theta_{u,j}$ and $\theta_{o,j}$ are the $j$-th components of the model weights. The entire term being minimized during removal is:

$$\mathcal{L}_{\text{forget}}(\mathcal{M}_{\boldsymbol{\theta}_u}; \mathcal{M}_{\boldsymbol{\theta}_o}, \mathcal{D}_f, \mathcal{D}_r) = \mathcal{L}_{\text{unlearn}}(\mathcal{M}_{\boldsymbol{\theta}_u}; \mathcal{D}_f) + \lambda \mathcal{L}_{\text{EWC}}(\mathcal{M}_{\boldsymbol{\theta}_u}; \mathcal{M}_{\boldsymbol{\theta}_o}, \boldsymbol{f}^{(\mathcal{D}_r)}, \boldsymbol{f}^{(\mathcal{D}_f)}) \quad (8)$$

where $\lambda \geq 0$ is a hyperparameter that balances the two objectives. While minimizing $\mathcal{L}_{\text{forget}}$ induces forgetting on $\mathcal{D}_f$, we observe, similar to Kurmanji et al. (2023), that performance on $\mathcal{D}_r$ degrades. To mitigate this, we minimize the same loss $\mathcal{L}_{repair}$ on retain data as SCRUB+R (Equation 3).

As in SCRUB+R, we find that optimizing both $\mathcal{L}_{\text{forget}}$ and $\mathcal{L}_{\text{repair}}$ jointly leads to instabilities due to the conflicting nature of the objectives. To remedy this, we minimize the objectives in an interleaved fashion as described in Section 2.1. This procedure is detailed in Algorithm 1. In all experiments we fix $k = 1$, which constitutes the most efficient choice. The observation that we only need to sample minibatches from $\mathcal{D}_r$ to maintain performance was key for the efficiency gains seen in Table 3. However, for larger datasets we suspect that setting $k > 1$ may be necessary to retain generalizability.

## 3.1 EVALUATION

We evaluate TA on the MNIST (Deng, 2012), CIFAR-10, CIFAR-100 (Krizhevsky, 2009), and Pins Face Recognition (Burak, 2019) datasets. The performance of the unlearned model ($\mathcal{M}_{\boldsymbol{\theta}_u}$) is benchmarked against a **retrained model**, which is trained from scratch on only the retain set, $\mathcal{D}_r$. This retrained model represents the gold standard for unlearning.

We assess performance across three key dimensions:

- **Model Utility**: We measure accuracy on the test set to ensure the model's performance on retained knowledge is not degraded. The utility of the unlearned model should remain comparable to that of the retrained model.

- **Unlearning Efficacy**: To confirm information removal, we measure the unlearned model's accuracy on the forget set $\mathcal{D}_f$. Effective unlearning is achieved when this accuracy drops to the level of the retrained model.

- **Privacy**: A model's unusually high error rate on specific data points can signal to an attacker that they were part of a forget set. To quantify this vulnerability, we measure the model's exposure to Membership Inference Attacks (MIA) (Shokri et al., 2017), following the implementation from Foster et al. (2024).

Crucially, the unlearned model should remain cross to retraining across these criteria. For instance, when forgetting an entire class, it is not sufficient to have perfect unlearning efficacy at the cost of catastrophically forgetting retain or validation data.

### 3.1.1 CONVERGENCE

Our MNIST experiments are designed to highlight the instabilities in SCRUB+R that motivated Teacher Ascent. To accelerate experimentation, we subsample the training set to $10,000$ images. Experiment details surrounding the model architecture, training parameters, and data processing are included in Appendix B.

Bertram et al. (2019) found that real-world removal requests come from a small subset of actors. To simulate this experimentally, we construct forget sets from local neighborhoods of a t-SNE embedding space (Maaten & Hinton, 2008). We define three forget sets from rectangular t-SNE regions with varying class compositions, as detailed in Table 1.

Table 1: Different retain/forget splits based on the t-SNE qualitative selection, along with their forget set class distributions.

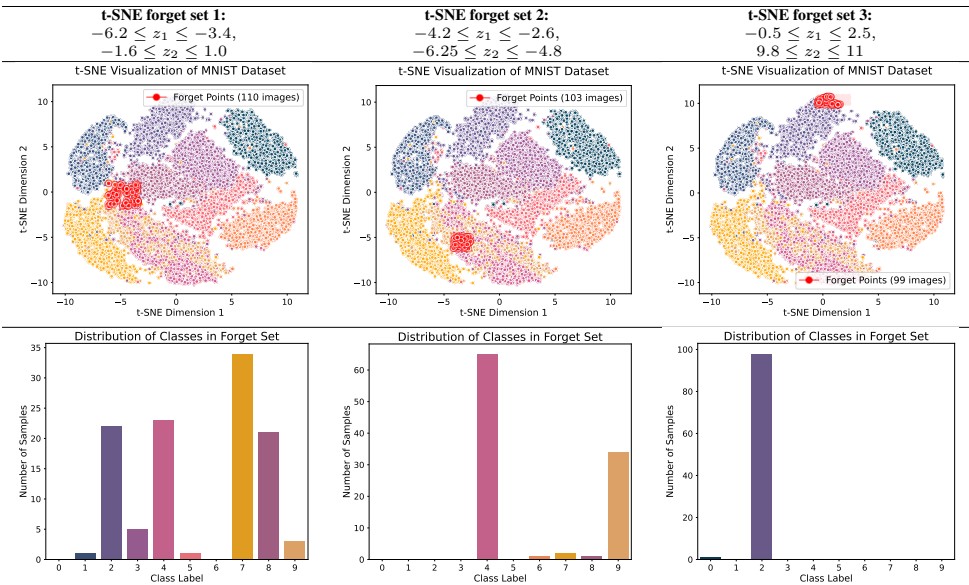

### 3.1.2 BENCHMARKING

The goal of this experiment is to evaluate TA against other established MU methods on a variety of unlearning tasks. To this end, experiments on CIFAR-10, CIFAR-100, and Pins Face Recognition are carried out. Across these, a vision transformer (Dosovitskiy et al., 2021) with a classification head is used, and all parameters are optimized during model trainings. Details on the specific architecture, training configuration, and data preprocessing are provided in Appendix B. For all datasets, we consider forgetting an entire class. For CIFAR-10, we conduct additional experiments with forgetting mislabeled data and subsets of a class. These scenarios and their motivation are outlined below:

**Forgetting a Class:** A common baseline for assessing unlearning effectiveness. This mimics a scenario where one has to remove sensitive knowledge from a model, such as dangerous information, explicit content, or copyrighted material.

**Forgetting a random subset of a class:** In this setting, there is naturally some information overlap between the retain and forget data. This resembles a situation where a deletion request has been made for observations that a retrained model can, to some extent, generalize to.

**Forgetting Corrupted Data:** Removing a small set of mislabeled samples to test the model's ability to correct data contamination, a common issue in real-world datasets. It has been shown that some of the most common datasets have at least $3.3\%$ mislabeled samples (Northcutt et al., 2021).

A common paradigm in unlearning evaluation is to search for hyperparameters such that the unlearned model is as close to a retrained model as possible. This, however, does not resemble a practical unlearning setting where one cannot determine these optimally. To provide a more realistic and fair benchmark, we introduce, to our knowledge, a new evaluation protocol: for each scenario,

we select hyperparameters by optimizing performance on a separate but semantically related proxy unlearning task. The best hyperparameters from the proxy task are then used, without modification, for the final downstream evaluation. This setup is detailed in Table 2. We conduct the hyperparameter search using Optuna's Tree-structured Parzen Estimator (TPE) (Bergstra et al., 2011; Akiba et al., 2019). For each proxy task, we run 30 trials and apply the best-performing hyperparameter configuration to its corresponding downstream task.

Table 2: Forget set construction strategy on the various benchmarks for the downstream task as well as hyperparameter search.

| Dataset | Forget set type | Downstream forget set | Hyperparameter search forget set |
|---------|-----------------|----------------------|----------------------------------|
| CIFAR-10 | Whole class | Forget all images in the ship class. | Forget all images in the airplane class. |
| CIFAR-10 | Subclass | Forget 500 samples (10%) from the horse class. | Forget 500 samples (10%) from the deer class. |
| CIFAR-10 | Corrupted | Forget 200 samples from the automobile class that were mislabeled as belonging to the truck class. | Forget 200 samples from the airplane class that were mislabeled as belonging to the boat class. |
| CIFAR-100 | Whole class | Forget all images in the rocket class | Forget all images in the bridge class. |
| Pins FR | Whole class | Forget all images (173) of Tom Cruise | Forget all images (110) of Zac Efron |

Note that we generally pick the hyperparameter search forget set to belong to the same super-class as the downstream forget set e.g., when forgetting an entire class in CIFAR-10, both forget sets contain vehicles. We deviate from this only on the CIFAR-100 task to gauge the effect of increasing the dissimilarity between the downstream and hyperparameter search forget sets.

## 4 RESULTS AND DISCUSSION

First, we investigate the convergence of SCRUB+R and TA on the MNIST forget sets seen in Table 1. To assess how the number of total rounds and forget rounds affect the performance of the two methods, we compute the model accuracies as a function of total rounds in Figure 1. We set the number of forgetting rounds to $R_f = \frac{R}{2}$ for both SCRUB+R and TA.

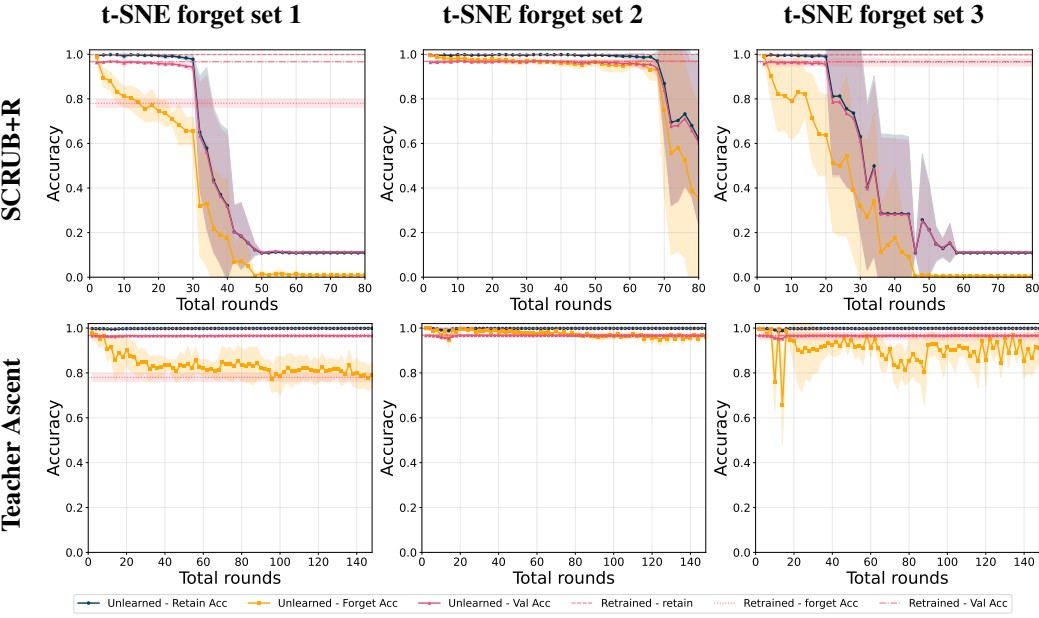

Figure 1: Accuracies of the unlearned model after applying SCRUB+R and TA as a function of the total number of rounds. Mean and std. of the accuracies are provided and were computed over 5 seeds with different model initializations. Results are reported for the three MNIST forget sets in Table 1. This illustrates that while SCRUB+R drops the forget set accuracy, the model eventually suffers from catastrophic forgetting. This instability, and its dependence on the specific forget set $\mathcal{D}_f$, complicates hyperparameter selection, particularly the number of unlearning rounds.

As evident from Figure 1, the unlearned models produced by SCRUB+R are highly dependent on the number of forget rounds. We fix the hyperparameters in this experiment (see Appendix B), however, the onset of catastrophic forgetting in SCRUB+R was observed consistently irrespective of these. In Figure 1, only a narrow range of the unlearned models on t-SNE forget set 1, those around 10-20 total rounds, approach the forget accuracy of a retrained model. Meanwhile on t-SNE forget set 3, choosing exactly 2 and 4 total rounds are the **only** configurations that approaches a retrained model. Looking at the resulting unlearned models for the three forget sets in combination, it is clear that there is no trivial way of pre-determining the appropriate number of forget and repair epochs. Meanwhile, the unlearned models produced by TA, as seen in Figure 1, match a retrained model on the forget set far more consistently. Furthermore, the retain and validation accuracies are unaffected by the number of rounds, addressing a key limitation of SCRUB+R.

To give insight into the dynamics of TA during unlearning as well as assess whether protecting parameters important for the retain set affects the unlearned model, we plot accuracies as a function of rounds for varying $\lambda$ in Figure 2. As seen, there is little deviation between the final unlearned model at round 100 for $\lambda \in \{2, 64\}$. However, omitting regularization entirely significantly degrades the unlearned model's performance on $\mathcal{D}_f$. We report further results on this in Appendix C. Herein, it also appears that the variability of the final unlearned models' forget accuracy increases when omitting EWC regularization.

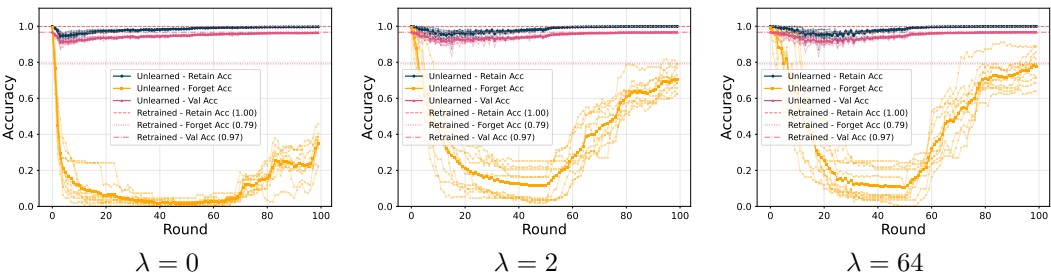

$$\lambda = 0 \qquad\qquad \lambda = 2 \qquad\qquad \lambda = 64$$

Figure 2: Effect of different regularization strengths on the convergence of Teacher Ascent. Experiments were run on t-SNE forget set 1 for 100 total rounds.

Next, we benchmark TA against SCRUB+R, SSD, and LoTUS, seen in Table 3, on the tasks unlearning tasks listed in Table 2. For reference, we also report the performance of the original model. Note that LoTUS is omitted from the corrupted task since the method is ill defined for that type of unlearning setting. Considering the CIFAR-10 unlearning tasks, TA emerges as the most viable method when factoring in computation time. Its accuracy across data splits consistently tracks that of a retrained model while remaining 6 to 19 times faster than retraining. SCRUB+R also closely matches the retrained model, with the exception of forgetting mislabeled data. The main drawback with SCRUB+R lies with its efficiency, nearly matching that of a retrained model on most forget sets. The SSD method, while being efficient, is highly unreliable. When forgetting an entire class and a subset of a class, it remains too conservative and on mislabeled data the model performance degrades to random guessing. This highlights that the SSD hyperparameters are highly sensitive to the forget set and original model's learned representation. Meanwhile, LoTUS degrades retain and validation performances slightly on CIFAR-100 and Pins FR, and remains too conservative on subclass forgetting in CIFAR-10. The large differences in runtime for LoTUS can be attributed to the subset size hyperparameter being tuned which specifies how big a fraction of retain data should be used during unlearning. Also, the large standard deviation in runtime for LoTUS on the CIFAR-10 sub-class task is due to the method's early stopping mechanism.

In terms of privacy preservation, measured by the MIA probability, no unlearning method matches retraining exactly across forget sets. However, both TA and SCRUB+R yield a notable shift from the original model's privacy profile, suggesting that the unlearned model's relative uncertainty on the forget set increases and comes closer to resembling retraining. In the corrupted setting, the decreased MIA probabilities indicate that remnants of the mislabeled data still persist, resulting in higher uncertainties. It should be mentioned, however, that MIA measures have met some critique (Rezaei & Liu, 2021; Zhang et al., 2025). While TA and SCRUB+R show promise and lessen the gap in MIA probability to a retrained model, the remaining difference indicates that perfectly replicating the privacy profile of a retrained model is a challenging task and warrants further investigation.

Table 3: Accuracies on the benchmarks and unlearning tasks described in 3.1.2. The mean and standard deviation across 10 runs with different model initializations is reported. The forget set was kept constant across the different runs. Note that accuracies in the corrupted setting are reported for the true class. The method closest to retraining is highlighted in bold. Methods highlighted in gray failed in the unlearning setting, defined as having an absolute deviation of more than 20 percentage points from retraining on any accuracy metric. For runtime, the fastest method is highlighted in bold granted that it did not fail the unlearning task.

| Dataset | Forget set type | Method | MIA | Retain acc | Forget acc | Val acc | Time (sec) |
|---------|-----------------|--------|-----|------------|------------|---------|------------|
| CIFAR-10 | Whole class | Original | $0.97 \pm 0.01$ | $1.00 \pm 0.00$ | $1.00 \pm 0.00$ | $0.98 \pm 0.00$ | - |
| | | Retraining | $0.22 \pm 0.02$ | $1.00 \pm 0.00$ | $0.00 \pm 0.00$ | $0.88 \pm 0.00$ | $2466.99 \pm 25.27$ |
| | | SCRUB+R | $\mathbf{0.31 \pm 0.28}$ | $\mathbf{1.00 \pm 0.00}$ | $\mathbf{0.00 \pm 0.00}$ | $0.87 \pm 0.00$ | $2242.65 \pm 88.74$ |
| | | SSD | $0.82 \pm 0.31$ | $1.00 \pm 0.01$ | $0.91 \pm 0.27$ | $0.97 \pm 0.04$ | $71.87 \pm 0.21$ |
| | | LoTUS | $0.00 \pm 0.00$ | $0.98 \pm 0.00$ | $0.05 \pm 0.03$ | $0.85 \pm 0.00$ | $\mathbf{16.93 \pm 0.67}$ |
| | | TA | $0.00 \pm 0.00$ | $\mathbf{1.00 \pm 0.00}$ | $\mathbf{0.00 \pm 0.00}$ | $\mathbf{0.88 \pm 0.00}$ | $378.30 \pm 7.54$ |
| CIFAR-10 | Corrupted | Original | $0.17 \pm 0.04$ | $1.00 \pm 0.00$ | $0.16 \pm 0.05$ | $0.98 \pm 0.00$ | - |
| | | Retraining | $0.88 \pm 0.01$ | $1.00 \pm 0.00$ | $0.98 \pm 0.01$ | $0.98 \pm 0.00$ | $2932.59 \pm 75.44$ |
| | | SCRUB+R | $0.46 \pm 0.21$ | $0.96 \pm 0.03$ | $0.90 \pm 0.07$ | $0.93 \pm 0.03$ | $1602.98 \pm 87.19$ |
| | | SSD | $0.00 \pm 0.00$ | $0.10 \pm 0.02$ | $0.00 \pm 0.00$ | $0.10 \pm 0.02$ | $67.38 \pm 0.91$ |
| | | TA | $\mathbf{0.52 \pm 0.16}$ | $0.99 \pm 0.00$ | $\mathbf{1.00 \pm 0.01}$ | $\mathbf{0.97 \pm 0.00}$ | $153.03 \pm 3.10$ |
| CIFAR-10 | Subclass | Original | $0.98 \pm 0.01$ | $1.00 \pm 0.00$ | $1.00 \pm 0.00$ | $0.98 \pm 0.00$ | - |
| | | Retraining | $0.90 \pm 0.01$ | $1.00 \pm 0.00$ | $0.98 \pm 0.00$ | $0.98 \pm 0.00$ | $3012.37 \pm 112.59$ |
| | | SCRUB+R | $0.93 \pm 0.01$ | $\mathbf{1.00 \pm 0.00}$ | $\mathbf{0.98 \pm 0.01}$ | $\mathbf{0.98 \pm 0.00}$ | $2818.12 \pm 78.66$ |
| | | SSD | $0.94 \pm 0.06$ | $\mathbf{1.00 \pm 0.00}$ | $1.00 \pm 0.00$ | $\mathbf{0.98 \pm 0.00}$ | $\mathbf{65.89 \pm 1.22}$ |
| | | LoTUS | $0.92 \pm 0.05$ | $1.00 \pm 0.00$ | $1.00 \pm 0.01$ | $0.98 \pm 0.00$ | $649.36 \pm 237.81$ |
| | | TA | $\mathbf{0.89 \pm 0.09}$ | $0.99 \pm 0.01$ | $\mathbf{0.98 \pm 0.02}$ | $0.97 \pm 0.01$ | $172.16 \pm 0.08$ |
| CIFAR-100 | Whole class | Original | $0.93 \pm 0.01$ | $1.00 \pm 0.00$ | $1.00 \pm 0.00$ | $0.87 \pm 0.00$ | - |
| | | Retraining | $0.13 \pm 0.03$ | $1.00 \pm 0.00$ | $0.00 \pm 0.00$ | $0.86 \pm 0.00$ | $2710.57 \pm 135.72$ |
| | | SCRUB+R | $\mathbf{0.03 \pm 0.03}$ | $\mathbf{1.00 \pm 0.00}$ | $\mathbf{0.00 \pm 0.00}$ | $0.87 \pm 0.00$ | $2136.08 \pm 27.81$ |
| | | SSD | $0.01 \pm 0.00$ | $0.99 \pm 0.00$ | $\mathbf{0.00 \pm 0.00}$ | $0.85 \pm 0.00$ | $\mathbf{66.76 \pm 1.33}$ |
| | | LoTUS | $0.00 \pm 0.00$ | $0.96 \pm 0.00$ | $0.05 \pm 0.04$ | $0.78 \pm 0.00$ | $1035.39 \pm 0.42$ |
| | | TA | $0.02 \pm 0.03$ | $\mathbf{1.00 \pm 0.00}$ | $\mathbf{0.00 \pm 0.01}$ | $\mathbf{0.86 \pm 0.00}$ | $140.04 \pm 1.33$ |
| Pins FR | Whole class | Original | $0.81 \pm 0.04$ | $1.00 \pm 0.00$ | $1.00 \pm 0.00$ | $0.89 \pm 0.01$ | - |
| | | Retrained | $0.05 \pm 0.02$ | $1.00 \pm 0.00$ | $0.00 \pm 0.00$ | $0.88 \pm 0.01$ | $2654.86 \pm 23.26$ |
| | | SCRUB+R | $\mathbf{0.03 \pm 0.02}$ | $0.99 \pm 0.01$ | $\mathbf{0.00 \pm 0.00}$ | $0.83 \pm 0.01$ | $721.08 \pm 19.02$ |
| | | SSD | $0.01 \pm 0.01$ | $\mathbf{1.00 \pm 0.00}$ | $\mathbf{0.00 \pm 0.00}$ | $\mathbf{0.88 \pm 0.01}$ | $\mathbf{21.23 \pm 0.15}$ |
| | | LoTUS | $0.00 \pm 0.00$ | $1.00 \pm 0.00$ | $0.18 \pm 0.19$ | $0.89 \pm 0.01$ | $222.09 \pm 0.12$ |
| | | TA | $0.01 \pm 0.01$ | $\mathbf{1.00 \pm 0.00}$ | $\mathbf{0.00 \pm 0.00}$ | $\mathbf{0.88 \pm 0.01}$ | $147.19 \pm 0.57$ |

On CIFAR-100 and Pins Face Recognition, reported in Table 3, TA perfectly matches the retrained model across accuracies. Surprisingly, SSD performs consistently well on these benchmarks. This is impressive considering that the CIFAR-100 hyperparameter sweep forget set was from a different super-class than the downstream forget set. Perhaps, this can be attributed to having many classes and fewer samples per class resulting in the diagonal FIM being a better approximation of the full FIM. However, further investigation is required to verify this. It could be interesting to further investigate how well the various unlearn methods perform as the difference between the downstream forget set and the one used for hyperparameter search increases. We defer this to future research.

## 5 CONCLUSION

We propose Teacher Ascent, a novel unlearning method inspired by knowledge distillation and continual learning principles. Across different benchmarks and unlearning tasks, TA consistently tracks the behavior of a retrained model, shows less sensitivity to its hyperparameters, and remains highly efficient compared to retraining. By benchmarking the unlearning methods on suboptimal hyperparameters, the reported results are more faithful to real-life unlearning scenarios. The consistent results of TA in this setting represent a big step towards making unlearning viable in practical scenarios where one cannot search for ideal hyperparameters.

USE OF LLM STATEMENT

LLMs have been used for proofreading, writing code, and gaining an overview of the field of Machine Unlearning in the early stages of finding relevant work.

ETHICS STATEMENT

The authors declare no conflicts of interest.

REPRODUCIBILITY STATEMENT

All code for reproducing the experiments is available publicly at:
https://anonymous.4open.science/r/TeacherAscent-D065/

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

APPENDIX

## A PROPERTIES OF MAXIMIZING KL DIVERGENCE

**Setup.** Let the teacher and student produce temperature-scaled probability distributions

$$\boldsymbol{p} = \text{softmax}\Big(\frac{\boldsymbol{p}_{\text{logits}}}{\tau}\Big), \qquad \boldsymbol{q} = \text{softmax}\Big(\frac{\boldsymbol{q}_{\text{logits}}}{\tau}\Big),$$

where $\boldsymbol{p}_{\text{logits}} = \mathcal{M}_{\boldsymbol{\theta}_o}(\boldsymbol{x}) \in \mathbb{R}^C$ and $\boldsymbol{z} \triangleq \boldsymbol{q}_{\text{logits}} = \mathcal{M}_{\boldsymbol{\theta}_u}(\boldsymbol{x}) \in \mathbb{R}^C$. The unlearning objective being maximized in SCRUB+R is:

$$\boldsymbol{\theta}_u^* = \arg\max_{\boldsymbol{\theta}_u} D_{KL}(\boldsymbol{p}\|\boldsymbol{q}).$$

**Lemma 1** (Gradient wrt. student parameters)**.** *The gradient of the KL divergence with respect to student parameters $\boldsymbol{\theta}_u$ can be written as*

$$\frac{\partial D_{KL}(\boldsymbol{p}\|\boldsymbol{q})}{\partial \boldsymbol{\theta}_u} = \frac{1}{\tau}(\boldsymbol{q} - \boldsymbol{p})\frac{\partial \boldsymbol{z}}{\partial \boldsymbol{\theta}_u} \tag{9}$$

*Proof.* Starting from the loss and applying the chain rule:

$$\frac{\partial D_{KL}(\boldsymbol{p}\|\boldsymbol{q})}{\partial \boldsymbol{\theta}_u} = \frac{\partial D_{KL}(\boldsymbol{p}\|\boldsymbol{q})}{\partial \boldsymbol{q}}\frac{\partial \boldsymbol{q}}{\partial \boldsymbol{z}}\frac{\partial \boldsymbol{z}}{\partial \boldsymbol{\theta}_u}$$

The gradient wrt. student probabilities is given as:

$$\frac{\partial D_{KL}(\boldsymbol{p}\|\boldsymbol{q})}{\partial q_j} = \frac{\partial}{\partial q_j}\sum_{i=1}^{C} p_i \log \frac{p_i}{q_i} = -\frac{p_j}{q_j}, \qquad \frac{\partial D_{KL}(\boldsymbol{p}\|\boldsymbol{q})}{\partial \boldsymbol{q}} = \Big[-\frac{p_1}{q_1}, \quad \cdots, \quad -\frac{p_C}{q_C}\Big].$$

Next, we need to determine the Jacobian of $\boldsymbol{q}$ and we begin by finding its diagonal elements using the quotient rule:

$$\begin{aligned}
\frac{\partial q_i}{\partial z_i} &= \frac{\frac{1}{\tau}\exp(z_i/\tau)\sum_{j=1}^{C}\exp(z_j/\tau) - \exp(z_i/\tau)\frac{1}{\tau}\exp(z_i/\tau)}{\Big(\sum_{j=1}^{C}\exp(z_j/\tau)\Big)^2} \\
&= \frac{\frac{1}{\tau}\exp(z_i/\tau)\left(S - \exp(z_i/\tau)\right)}{S^2} \\
&= \frac{1}{\tau}\frac{\exp(z_i/\tau)}{S}\frac{S - \exp(z_i/\tau)}{S} \\
&= \frac{1}{\tau}q_i(1 - q_i)
\end{aligned}$$

And considering the case where we compute $\frac{\partial q_i}{\partial z_j}, i \neq j$ we get:

$$\begin{aligned}
\frac{\partial q_i}{\partial z_j} &= \frac{-\exp(z_i/\tau)\cdot\frac{1}{\tau}\exp(z_j/\tau)}{S^2} \\
&= -\frac{1}{\tau}\frac{\exp(z_i/\tau)}{S}\frac{\exp(z_j/\tau)}{S} \\
&= -\frac{1}{\tau}q_iq_j
\end{aligned}$$

Putting this together, we get that the Jacobian of $\boldsymbol{q}$ is:

$$\frac{\partial q_i}{\partial z_j} = \begin{cases} \frac{1}{\tau}q_i(1 - q_i), & i = j, \\ -\frac{1}{\tau}q_iq_j, & i \neq j, \end{cases} \qquad \frac{\partial \boldsymbol{q}}{\partial \boldsymbol{z}} = \frac{1}{\tau}\begin{bmatrix} q_1(1 - q_1) & -q_1q_2 & \cdots & -q_1q_C \\ -q_2q_1 & q_2(1 - q_2) & \cdots & -q_2q_C \\ \vdots & \vdots & \ddots & \vdots \\ -q_Cq_1 & -q_Cq_2 & \cdots & q_C(1 - q_C) \end{bmatrix}$$

Next, the product of the first two terms in the chain rule is determined. The $j$'th row is given as:

$$\frac{\partial D_{KL}(\boldsymbol{p}\|\boldsymbol{q})}{\partial \boldsymbol{q}}\frac{\partial \boldsymbol{q}}{\partial z_j} = \frac{1}{\tau}\sum_{i=1}^{C}-\frac{p_i}{q_i}q_i(\delta_{ij}-q_j) \tag{10}$$

$$= -\frac{1}{\tau}\sum_{i=1}^{C}p_i(\delta_{ij}-q_j) \tag{11}$$

$$= -\frac{1}{\tau}\left(\sum_{i=1}^{C}p_i\delta_{ij} - \sum_{i=1}^{C}p_iq_j\right) \tag{12}$$

$$= -\frac{1}{\tau}\left(p_j - q_j\sum_{i=1}^{C}p_i\right) \tag{13}$$

$$= -\frac{1}{\tau}(p_j - q_j) \tag{14}$$

$$= \frac{1}{\tau}(q_j - p_j) \tag{15}$$

$$\tag{16}$$

Thus, we arrive at:

$$\frac{\partial D_{KL}(\boldsymbol{p}\|\boldsymbol{q})}{\partial \boldsymbol{q}}\frac{\partial \boldsymbol{q}}{\partial \boldsymbol{z}} = \frac{1}{\tau}(\boldsymbol{q}-\boldsymbol{p})$$

$\square$

**Assumption 1** (Model activations have an unbounded codomain). *Assume that the activation function of $\mathcal{M}$ has an unbounded codomain. As a result, $\mathcal{M}$ can produce arbitrarily large logits.*

**Proposition 1** (Parameter norm diverges without bounding). *Maximizing $D_{KL}(\boldsymbol{p}\|\boldsymbol{q})$ without bounding components of $\boldsymbol{q}$ admits no finite critical point for logits $\boldsymbol{z}$. Consequently, gradient ascent causes the norm of the student parameters to diverge $\|\boldsymbol{\theta}_u\| \to \infty$.*

*Sketch.* From Lemma 1, the gradient at the logits level is proportional to $\boldsymbol{q}-\boldsymbol{p}$. For any class $j$ with $p_j > 0$, the KL objective pushes $q_j \to 0$, requiring $z_j \to -\infty$ or other logits $z_i \to +\infty$. Because $\boldsymbol{q}-\boldsymbol{p}$ has nonzero magnitude, this produces a constant gradient in parameter space. Gradient ascent on this signal increases $\|\boldsymbol{\theta}_u\|$ without bound. $\square$

—

**Definition 1** (Bounded student probabilities). *For $\epsilon > 0$, define*

$$\tilde{q}_i = \max(\epsilon, q_i). \tag{17}$$

**Proposition 2** (KL-divergence under bounded probabilities). *For student probabilities $\tilde{\boldsymbol{q}}$, bounded using Equation 17, the objective becomes upper bounded by:*

$$D_{KL}(\boldsymbol{p}\|\tilde{\boldsymbol{q}}) = \sum_i p_i \log\frac{p_i}{\tilde{q}_i} \le -H(\boldsymbol{p}) + \log\frac{1}{\epsilon}.$$

**Lemma 2** (Gradient with bounding). *Defining the set of active classes $\mathcal{A} = \{k \mid q_k > \epsilon\}$ and the corresponding teacher probability mass $P_{\mathcal{A}} = \sum_{k\in\mathcal{A}}p_k$. Then*

$$\frac{\partial D_{KL}(\boldsymbol{p}\|\tilde{\boldsymbol{q}})}{\partial z_j} = \begin{cases} \frac{1}{\tau}(q_j P_{\mathcal{A}} - p_j), & j \in \mathcal{A}, \\ \frac{1}{\tau}(q_j P_{\mathcal{A}}), & j \notin \mathcal{A}. \end{cases}$$

*Proof.* Let $\mathbb{1}(\cdot)$ denote the indicator function. We begin by computing the gradient of the loss with respect to unbounded student probabilities:

$$\frac{\partial D_{KL}(\boldsymbol{p}\|\tilde{\boldsymbol{q}})}{\partial q_j} = \begin{cases} -\frac{p_j}{q_j} & q_j > \epsilon \\ 0 & q_j \le \epsilon \end{cases} \qquad \frac{\partial D_{KL}(\boldsymbol{p}\|\tilde{\boldsymbol{q}})}{\partial \boldsymbol{q}} = \left[\mathbb{1}(q_1 \ge \epsilon)\frac{-p_1}{q_1}, \dots, \mathbb{1}(q_C \ge \epsilon)\frac{-p_C}{q_C}\right]$$

The product of the first two terms in the chain rule becomes:

$$
\begin{aligned}
\frac{\partial D_{KL}(\boldsymbol{p}\|\tilde{\boldsymbol{q}})}{\partial \boldsymbol{q}}\frac{\partial \boldsymbol{q}}{\partial z_j} &= \frac{1}{\tau}\sum_{i=1}^{C}\mathbb{1}(q_i \geq \epsilon)\frac{-p_i}{q_i}q_i(\delta_{ij}-q_j)\\
&= -\frac{1}{\tau}\sum_{i\in\mathcal{A}}\frac{p_i}{q_i}q_i(\delta_{ij}-q_j)\\
&= -\frac{1}{\tau}\sum_{i\in\mathcal{A}}p_i(\delta_{ij}-q_j)\\
&= -\frac{1}{\tau}\left(\sum_{i\in\mathcal{A}}p_i\delta_{ij}-\sum_{i\in\mathcal{A}}p_iq_j\right)\\
&= -\frac{1}{\tau}\left(p_j\mathbb{1}(j\in\mathcal{A})-q_j\sum_{i\in\mathcal{A}}p_i\right)\\
&= \frac{1}{\tau}(q_jP_{\mathcal{A}}-p_j\mathbb{1}(j\in\mathcal{A}))
\end{aligned}
$$

$\square$

**Corollary 1** (Gradient vanishing in the unlearning limit). *If the student suppresses all teacher-supported classes, i.e., $q_k \leq \epsilon$ for all $k$ where $p_k > 0$, then $P_{\mathcal{A}} \to 0$ and*

$$
\lim_{P_{\mathcal{A}}\to 0}\left\|\frac{\partial D_{KL}(\boldsymbol{p}\|\tilde{\boldsymbol{q}})}{\partial \boldsymbol{z}}\right\| = 0.
$$

*Remark.* Bounding introduces an automatic stopping mechanism: once student probabilities for teacher-supported classes fall below $\epsilon$, the effective gradient vanishes, preventing runaway parameter magnitudes.

## B  EXPERIMENT DETAILS

This section outlines additional implementation details surrounding the experiments. Generally, all experiments were carried out on a single V100 GPU and seeds were used for reproducibility.

**FIM ratio and numeric stability:** When computing the FIM ratio (Equation 7), zero or near-zero denominators can cause instabilities. For parameters with $f_j^{(\mathcal{D}_r)}/f_j^{(\mathcal{D}_f)}$ undefined due to $f_j^{(\mathcal{D}_f)} = 0$, we replace the ratio with the maximum observed value within that model layer. This corresponds to treating such parameters as highly protected, since they provide no information about the forget set and should not serve as "free variables" for absorbing forgetting updates. An alternative is to set weights for $0/0$ cases to 1, thereby leaving irrelevant parameters unconstrained; both choices are defensible, and we adopt the more restrictive option to enforce that forgetting occurs only through parameters implicated in the forget set. For near-zero denominators, we clip all ratios at $10^6$. This preserves relative importance rankings while preventing unbounded weights. In practice, results are not sensitive to the choice of upper bound, since the other terms in the loss remain bounded.

### B.1  CIFAR AND PINS FACE RECOGNITION

For all experiments on CIFAR-10, CIFAR-100, and Pins Face Recognition, we use a vision transformer (Dosovitskiy et al., 2021) pre-trained on ImageNet (Deng et al., 2009) with mean pooling and a single classification layer on top[2]. Each model was trained for 20 epochs and a batch size of 128. AdamW was used as the optimizer with a learning rate of $10^{-4}$, weight decay of $10^{-3}$, and a cosine annealing learning rate scheduler with a period of 20 epochs. The final model was chosen as the one with the highest accuracy on the validation set. This architecture and training configuration was kept constant for all original and retrained models.

---

[2]The specific instance of ViT model being used is the tiny variant found here:
https://huggingface.co/WinKawaks/vit-tiny-patch16-224

For all experiments, we preprocess the images by resizing them to $224 \times 224$ using bilinear interpolation, re-scale the pixels to $[0, 1]$ by performing element-wise division with $255$. Hereafter, we normalizing them using the channel-wise mean and standard deviation of the training set, and converting labels to one-hot vectors.

For CIFAR-10 and CIFAR-100 we apply the CIFAR-10 AutoAugment policy for data augmentation described in Cubuk et al. (2019). For Pins Face Recognition, we use the following sequence of random augmentations: resized cropping, horizontal flipping with a 50% probability, a random rotation in the interval $[-10°, 10°]$, color jitter, and erasure. For all datasets, we apply channel-wise normalization after augmenting the training images. Augmentations are only performed during the training of the original and retrained models. Hence, at unlearning time the only transformation being applied to the training and retain datasets is normalization.

### B.1.1 Hyperparameter sweeps

To select hyperparameters, we use Optuna and run 30 trials for each unlearning method per forget set. An overview of the forget sets constructed for hyperparameter search as well as the downstream forget set can be found in Table 2.

We generally keep the upper and lower bounds of hyperparameters fixed for unlearning method and task in Table 2. One exception to this is with the number of total rounds for Teacher Ascent. Since the number of optimization steps scales with the size of the forget set, we change the bounds on the number of total rounds to have a similar number of total optimization steps for each unlearning task. During hyperparameter search, we seek to find a model that matches the retrained model accuracy on the forget and retain set as closely as possible.

### B.1.2 Corrupted Data

To avoid any confusion, we detail the exact procedure used for unlearning mislabeled data. We first draw 200 points from the automobile class and mislabel them as a truck. Hereafter, 10 original models are trained on the entire dataset including mislabeled data followed by 10 retrained models on the retain data (excluding mislabeled points entirely). Hereafter, unlearning is applied "as normal" e.g., no information about the data being mislabeled nor which class it actually belongs to.

During hyperparameter selection, the following is done: We draw 200 points from the boat class and mislabel them as a plane. We then train a single original model on the entire dataset including mislabeled data and a single retrained model on the retain dataset only. When unlearning, the forget loader still contains the corrupted labels e.g., when calculating the FIM as well as optimizing any objectives iterating over the forget set. When calculating the objective function for the hyperparameter sweep, the true labels are used for the forget set.

### B.2 MNIST

All original and retrained models on MNIST have the same architecture and model parameters. We use a neural network with 3 hidden layers, a hidden dimension of 3136, and residual skip connections between hidden layers. It was optimized using cross-entropy with Adam where we set weight decay to 0 a set learning rate of $10^{-3}$. Each model was trained for 30 epochs and the final model was chosen as the one with the highest validation accuracy. As part of data preprocessing, we min-max normalize the pixel values using the mean and standard deviation of the training set. Additionally, each $28 \times 28$ pixel image is flattened into a 784-dimensional vector. Lastly, the integer labels are converted into 10-dimensional one-hot encoded vectors.

For the MNIST experiments, we deliberately decided not to perform an expensive hyperparameter search. Rather, we specified sensible parameters and repeated the experiments to gauge the consistency of the results. This was chosen to resemble a practical unlearning scenario where one has a general idea about what parameters might be reasonable.

Specifically, we chose to run Teacher Ascent with $50\%$ of the total epochs containing the maximization step. The learning rate was set to $10^{-3}$, the same as when training the original model. For the distilled KL temperatures, we set $\tau_f = \tau_e = 2$ and $\tau_r = 5$.

For SCRUB+R, we set the temperatures to $\tau_f = \tau_r = 2$, use a learning rate of $10^{-3}$, and regularization strengths to $\alpha = \gamma = 2$.

The hyperparameters of SCRUB+R and TA were kept constant for all of the results provided on MNIST.

## C  EXTENDED MNIST RESULTS

Here we report additional results on the MNIST dataset. Each plots show the trajectory of 10 runs of Teacher Ascent during unlearning with different model seeds. The lines corresponding to a retrained model are the mean of the 10 retrained models across seeds. The three forget sets, seen in Table 1, were held constant.

### C.1  T-SNE FORGET SET 1

Table 4: Accuracies, MIA probabilites and Jensen-Shannon divergences for Teacher Ascent when run on t-SNE forget set 1 for 50 total rounds.

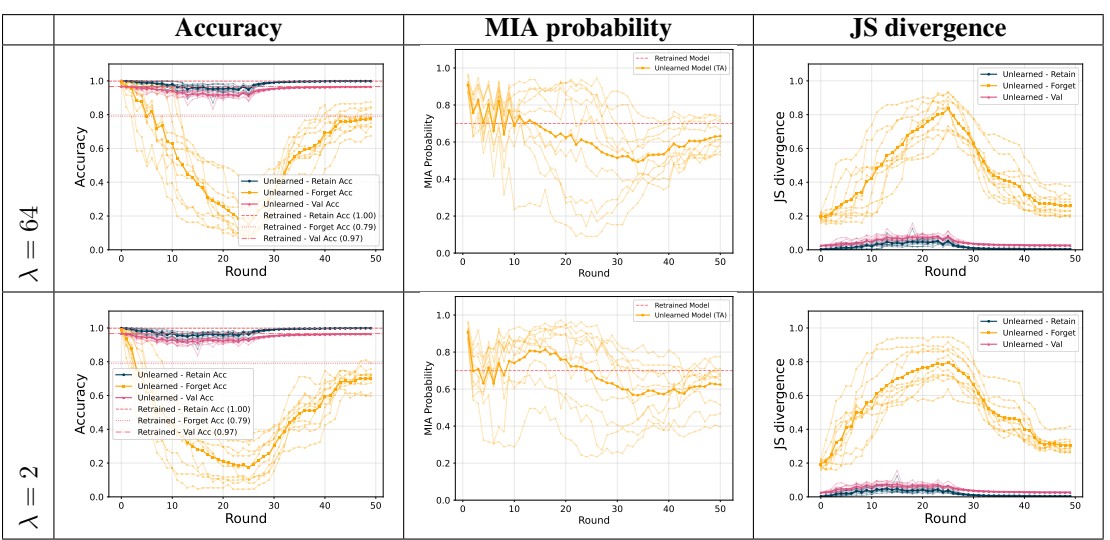

Table 5: Accuracies, MIA probabilites and Jensen-Shannon divergences for Teacher Ascent when run on t-SNE forget set 1 for 100 total rounds.

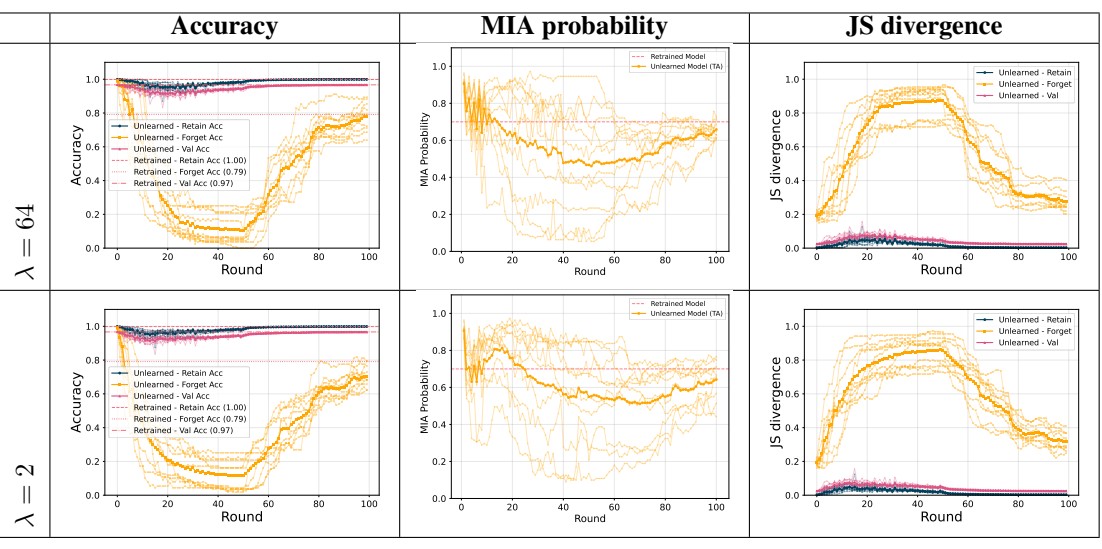

Table 6: Accuracies, MIA probabilites and Jensen-Shannon divergences for Teacher Ascent when run on t-SNE forget set 1 for 150 total rounds.

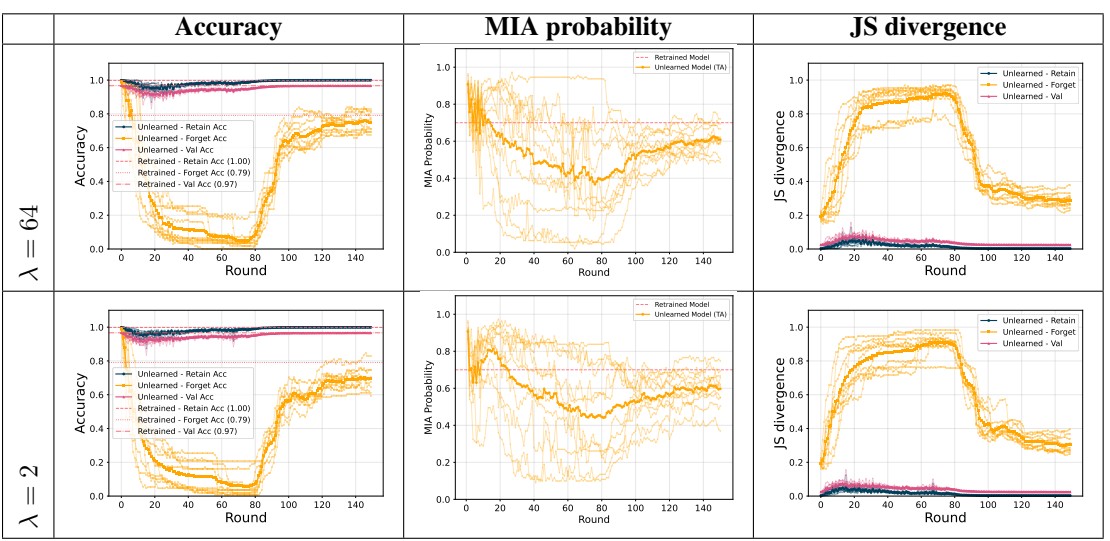

## C.2 T-SNE FORGET SET 2

Table 7: Accuracies, MIA probabilites and Jensen-Shannon divergences for Teacher Ascent when run on t-SNE forget set 2 for 50 total rounds.

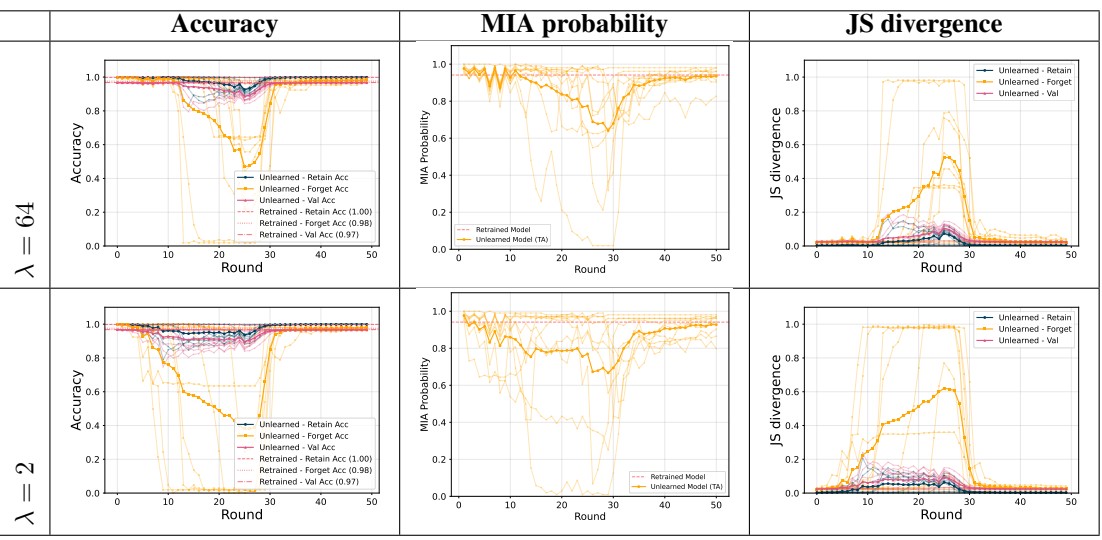

Table 8: Accuracies, MIA probabilites and Jensen-Shannon divergences for Teacher Ascent when run on t-SNE forget set 2 for 100 total rounds.

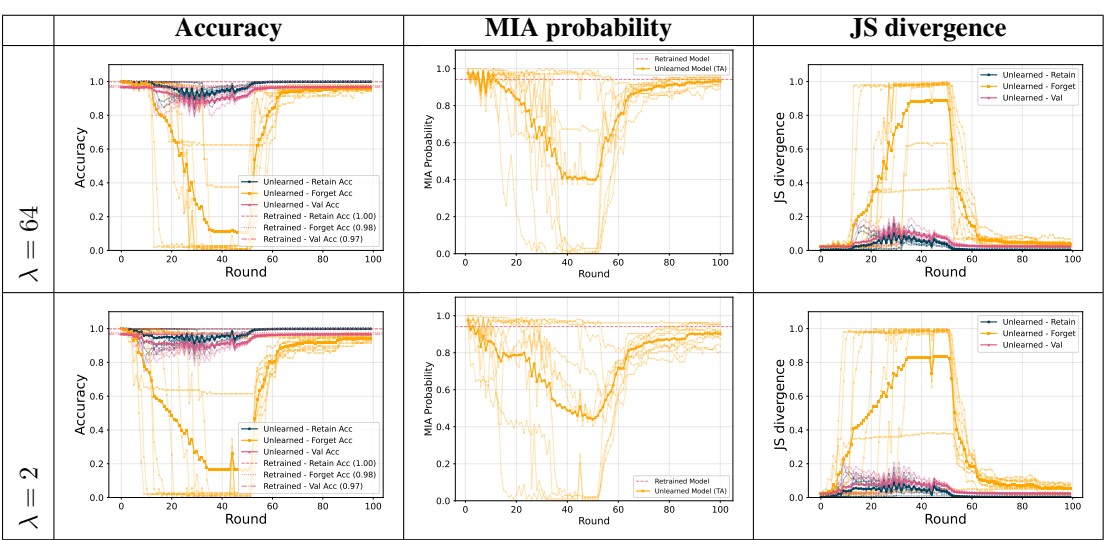

Table 9: Accuracies, MIA probabilites and Jensen-Shannon divergences for Teacher Ascent when run on t-SNE forget set 2 for 150 total rounds.

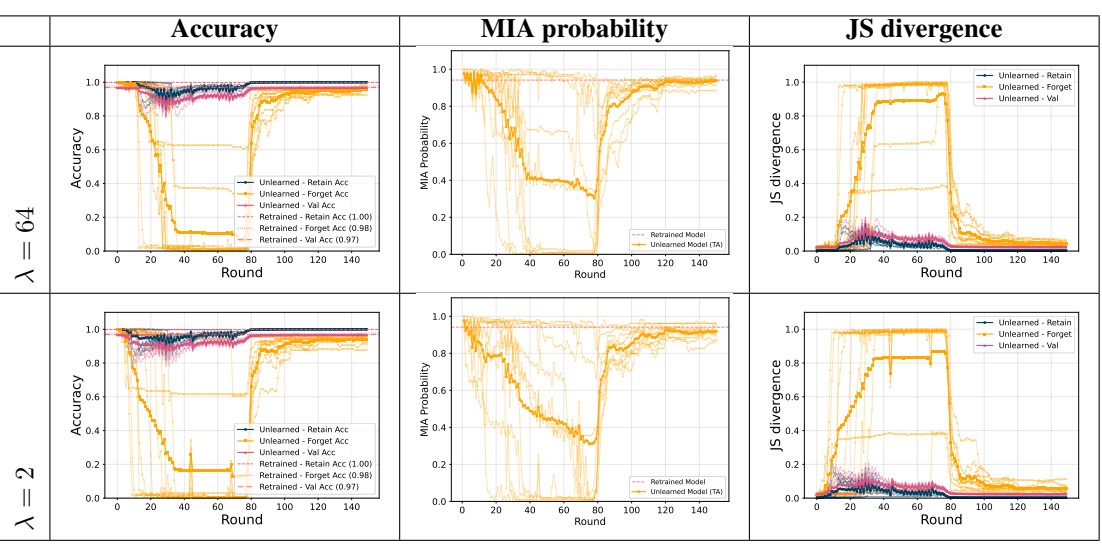

## C.3 T-SNE FORGET SET 3

Table 10: Accuracies, MIA probabilites and Jensen-Shannon divergences for Teacher Ascent when run on t-SNE forget set 3 for 50 total rounds.

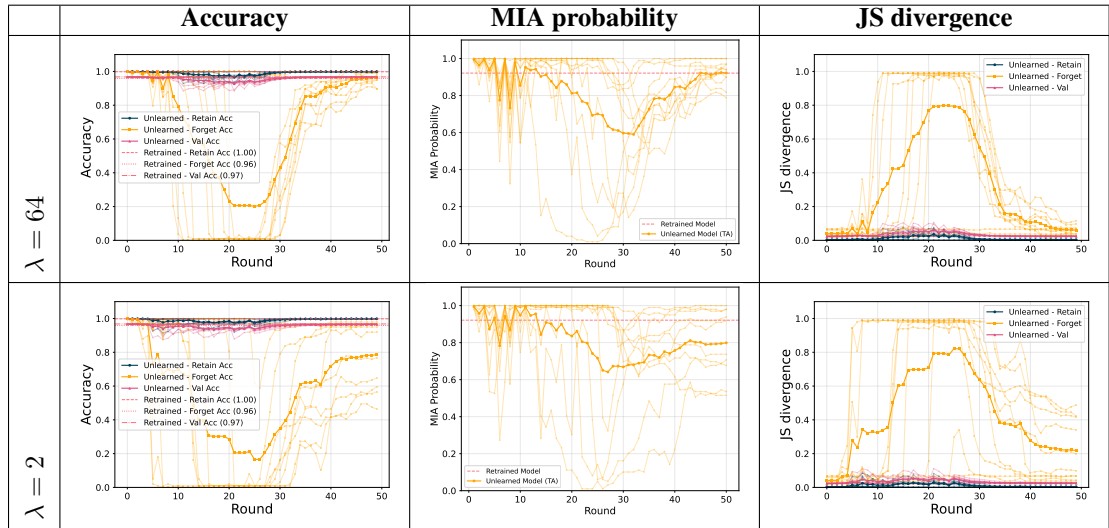

Table 11: Accuracies, MIA probabilites and Jensen-Shannon divergences for Teacher Ascent when run on t-SNE forget set 3 for 100 total rounds.

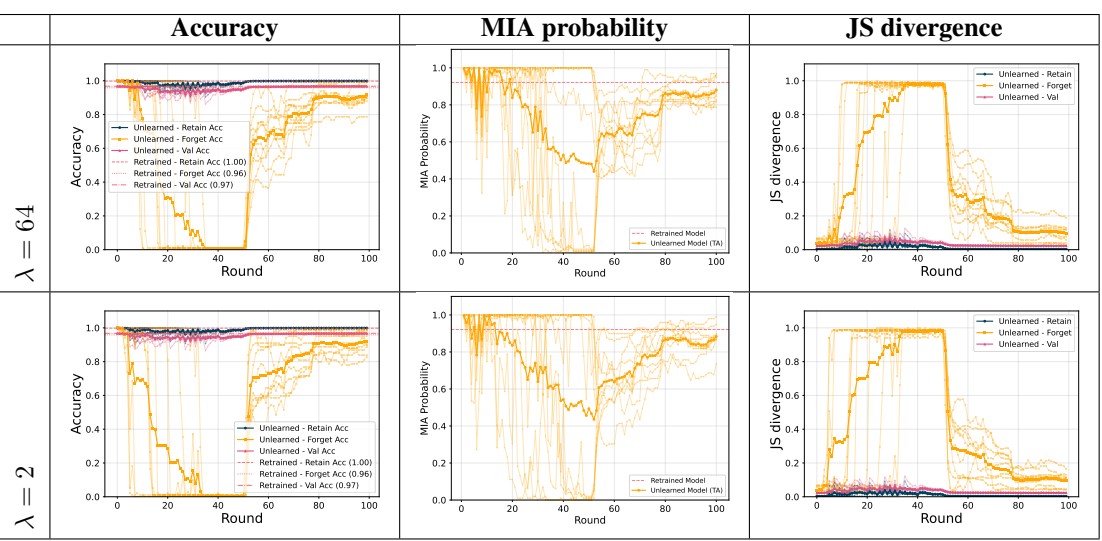

Table 12: Accuracies, MIA probabilites and Jensen-Shannon divergences for Teacher Ascent when run on t-SNE forget set 3 for 150 total rounds.

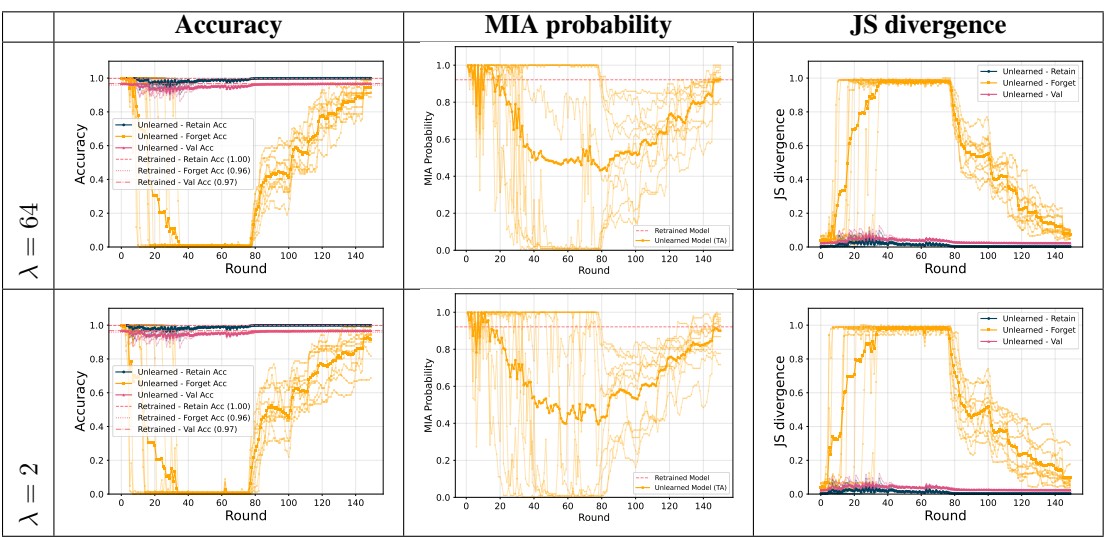

## C.4 ABLATION: THE EFFECT OF PROTECTING GENERALIZATION

Here we assess whether regularizing with the Fisher Information Matrix during the maximization step is beneficial.

Table 13: Accuracies, MIA probabilities, and Jensen-Shannon divergences on t-SNE forget set 1 when setting the regularization strength to $\lambda = 0$.

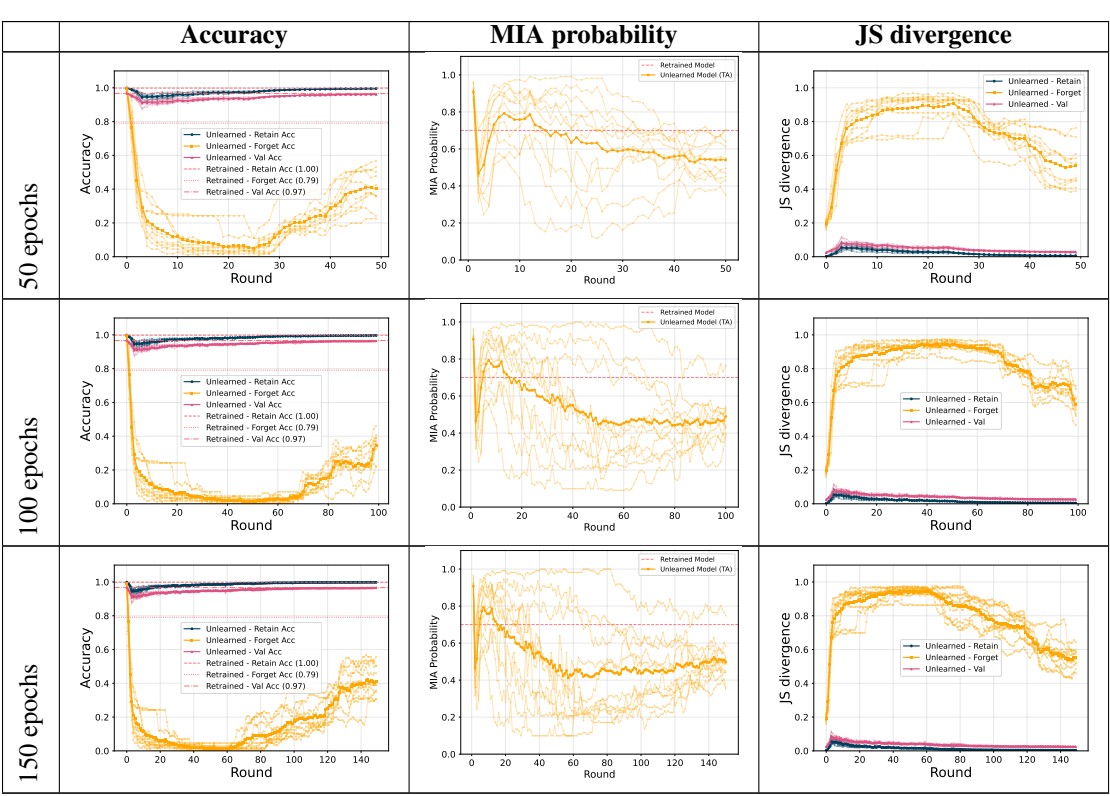

Table 14: Accuracies, MIA probabilities, and Jensen-Shannon divergences on t-SNE forget set 2 when setting the regularization strength to $\lambda = 0$.

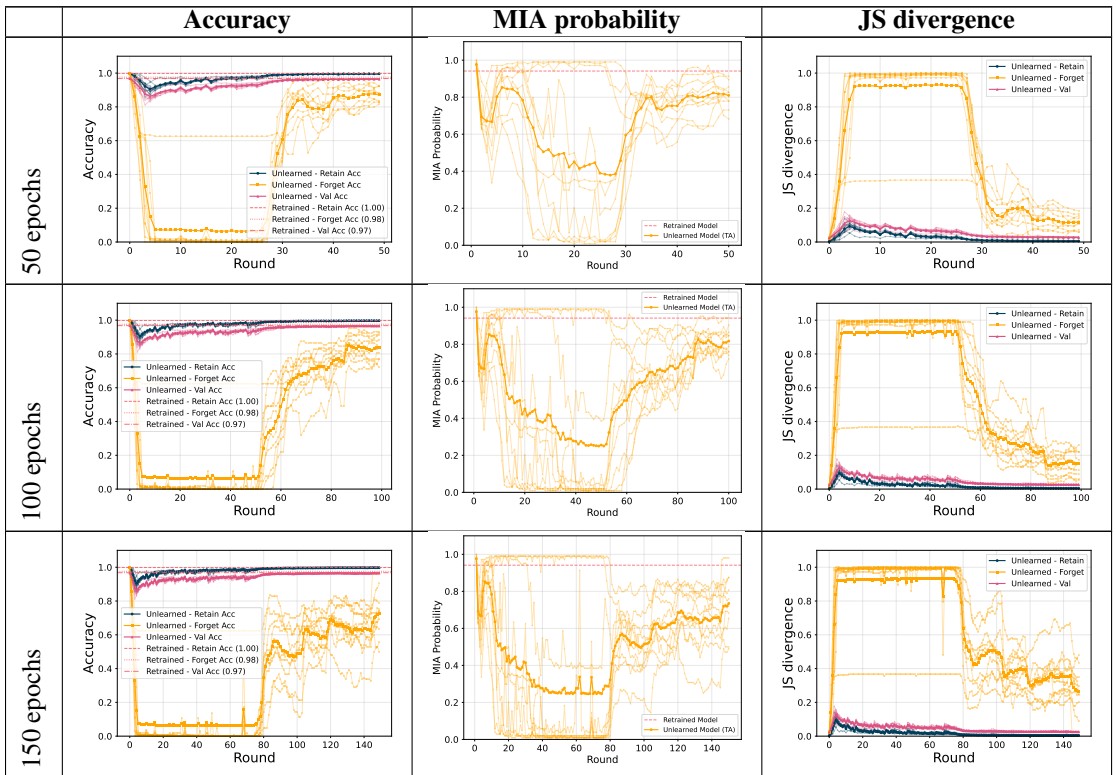

Table 15: Accuracies, MIA probabilities, and Jensen-Shannon divergences on t-SNE forget set 3 when setting the regularization strength to $\lambda = 0$.

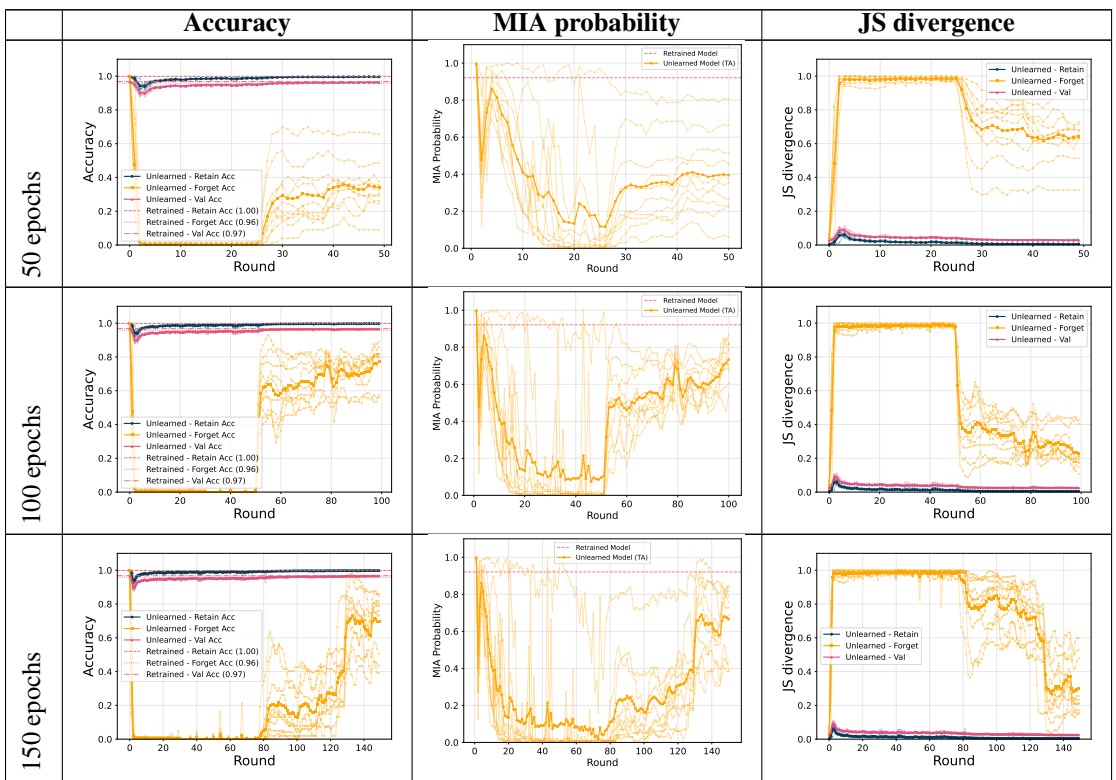

# D    CONTROLLED SETTING EXPERIMENTS

Here we report some additional results of Teacher Ascent in a controlled setting with three linearly separable classes. A small neural network was used, of similar architecture to the one in MNIST. All hyperparameters were kept the same as in the MNIST experiments except setting $\tau_e = 0.01$. We found this slightly improved performance in adversarial settings (scenarios 3 and 5) but setting $\tau_e = 2$ still performed comparatively. We suspect that shrinking the influence of the entropy term works well in these settings is due to the original model being highly uncertain on forget points.

Table 16: Controlled setting results for unlearning scenario 1.

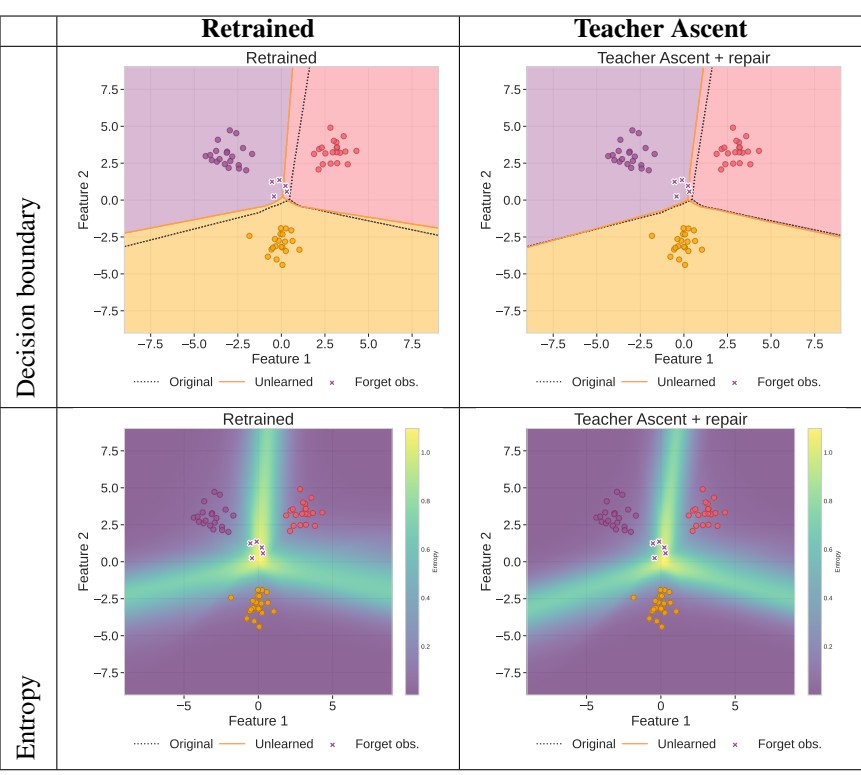

Table 17: Controlled setting results for unlearning scenario 2.

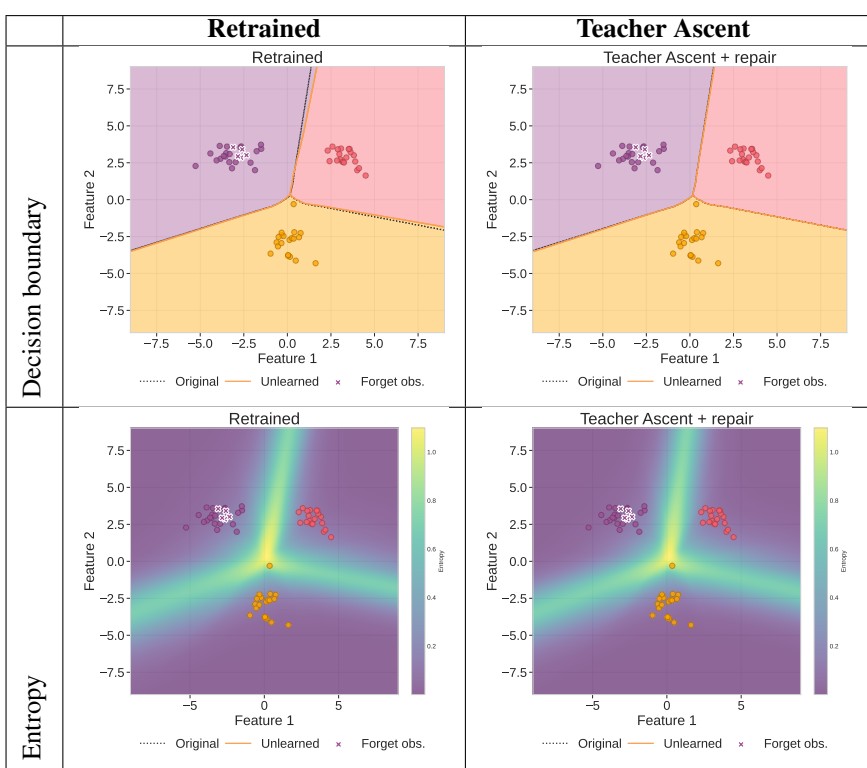

Table 18: Controlled setting results for unlearning scenario 3.

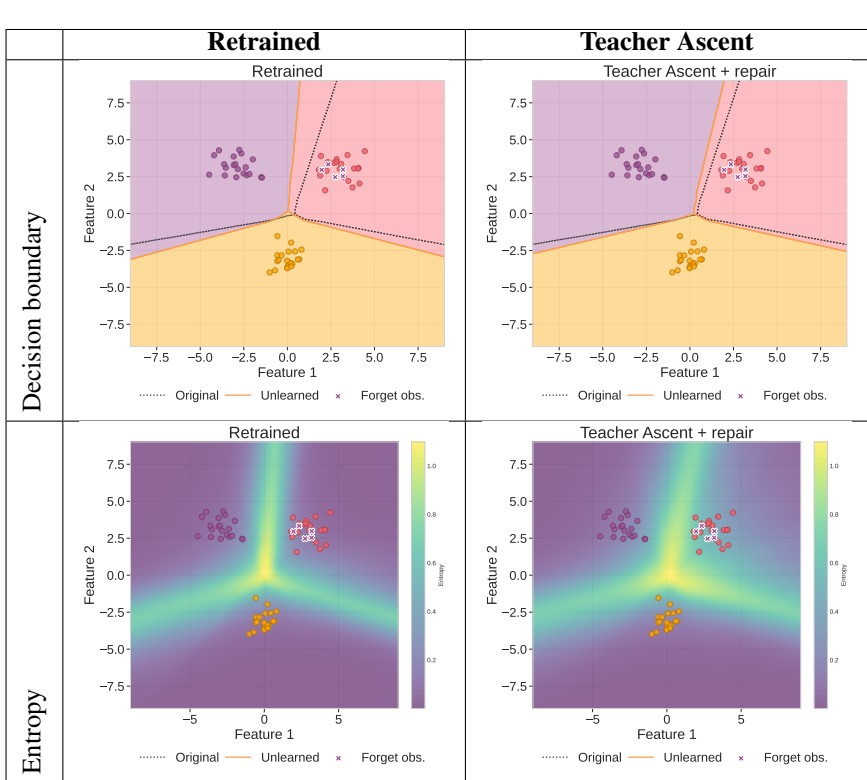

Table 19: Controlled setting results for unlearning scenario 4.

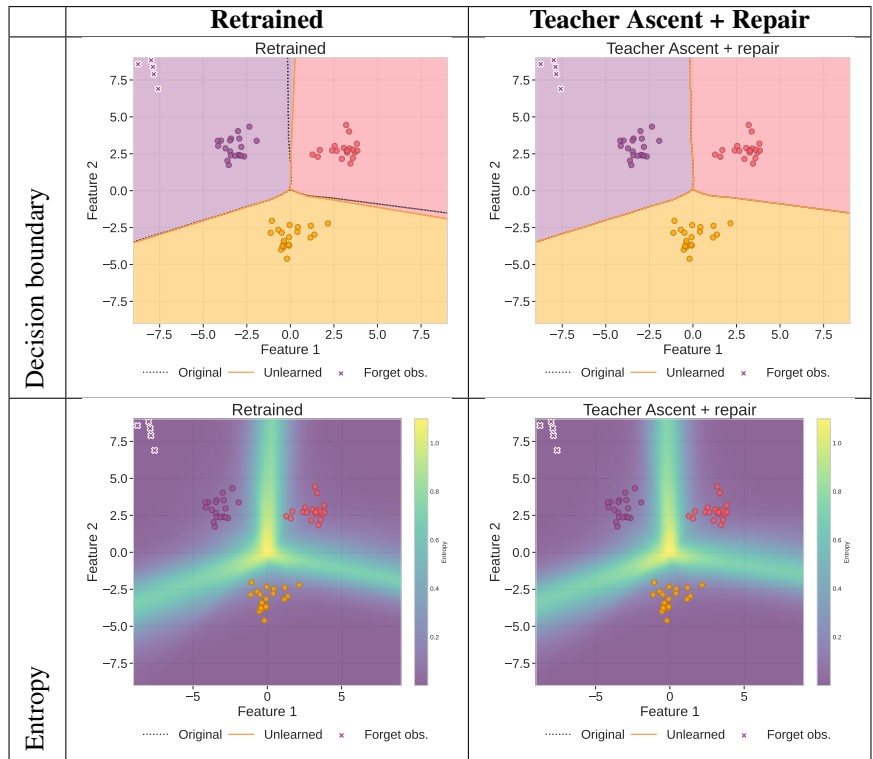

Table 20: Controlled setting results for unlearning scenario 5.

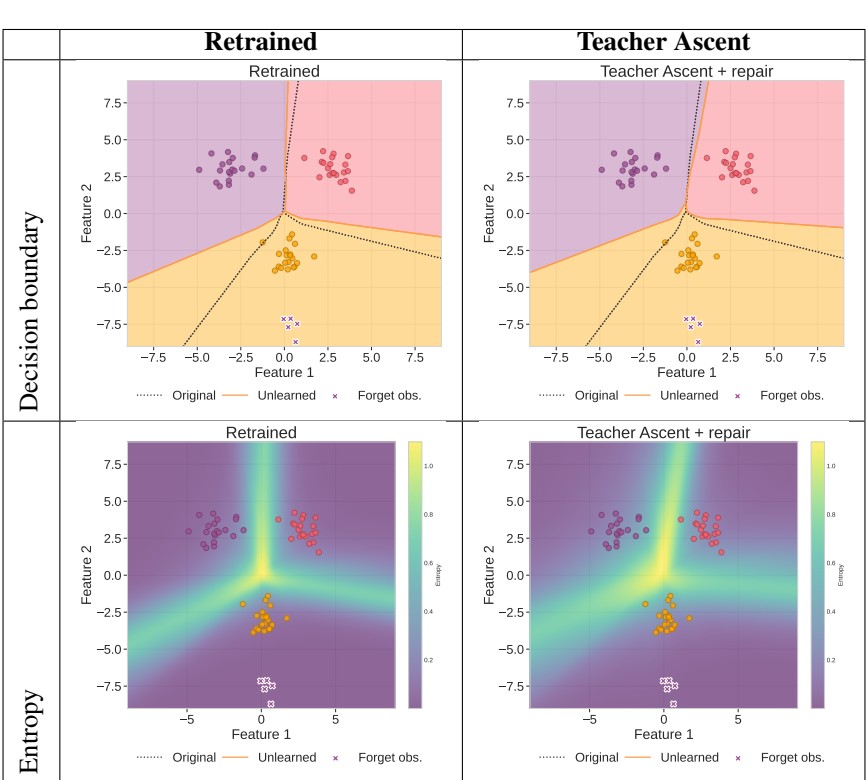

