# OpenReview forum: "Teacher Ascent: Robust and Efficient Machine Unlearning via Knowledge Distillation and Continual Learning"
_ICLR.cc/2026/Conference — Submitted to ICLR 2026_

### Official Review · Reviewer_PXQ3 · 2025-10-28

**Soundness:** 2
**Presentation:** 2
**Contribution:** 2
**Rating:** 4
**Confidence:** 4

**Summary:**

The paper introduces Teacher Ascent(TA), a novel and efficient fine-tuning method for Machine Unlearning designed to overcome the critical hyperparameter sensitivity and stability issues, such as catastrophic forgetting, inherent in prior state-of-the-art methods like SCRUB+R and SSD. Teacher Ascent is built on principles from knowledge distillation and continual learning, aiming to remove the influence of a specified forget set ($D_f$) while preserving performance on the retain set ($D_r$), thereby producing a model functionally equivalent to one trained only on $D_r$. The method achieves this stability through two key mechanisms: maximizing bounded Kullback-Leibler divergence terms during data removal to circumvent catastrophic forgetting, and employing a regularization term inspired by Elastic Weight Consolidation (EWC) that uses the Fisher Information Matrix (FIM) to protect parameters essential for generalization. Across diverse unlearning scenarios involving entire classes, subclasses, and mislabeled samples on datasets like MNIST, CIFAR, and Pins Face Recognition, the results demonstrate that Teacher Ascent consistently mimics the functional behavior of a retrained model.

**Strengths:**

- TA effectively mitigates catastrophic forgetting and demonstrates robustness across a wide range of hyperparameters, solving a critical reliability gap present in state-of-the-art methods. Specifically, TA achieves this stability by employing bounded KL divergence terms during the removal of the forget set, and incorporating a regularization term (inspired by EWC) that protects parameters essential for generalization using the Fisher Information Matrix (FIM) ratio of the retain and forget sets.

- TA is significantly more efficient than the full retraining from scratch on retain sets. Across various unlearning tasks, TA is shown to be 6 to 19 times more efficient than retraining while still matching the performance of the retrained model.

**Weaknesses:**

- The comparison currently centers on SSD and SCRUB+R, but several contemporary KD-based unlearning methods use the original model as a teacher and thus provide a natural point of reference for TA (e.g., DELETE[1]: decoupled/masked distillation). Even if some assumptions differ, including at least one such baseline—or clearly stating exclusion criteria (data access, objective, or compute)—would make the empirical section more complete.

- TA targets a full-access scenario where both $D_r$ and $D_f$ are available at unlearning time, and the method explicitly relies on $D_r$. Considering the realistic scenario in which an unlearning request comes in from the outside, such full-access assumption is not suitable for unlearning. The paper reveals that only mini-batch is sampled from $D_r$ in the Repair stage, fixed to k=1 in all experiments, and only speculates that "a large dataset may require k>1". There is no experiment/ablation to systematically reduce (ratio and k change) the $D_r$ usage or evaluate the 'only partially used' variation.

- In Table 3, please clarify the boldface rule for Time. If SSD (or any other method) is faster but fails the unlearning criteria, consider (a) bolding the fastest among methods that meet success criteria, with those criteria stated in the caption, and (b) visually flagging failed settings (e.g., gray or a footnote). This prevents misinterpretation of boldface as the absolute column minimum.

- Line 86, Minor typos: “Elastic Weight Consollidation” → “Consolidation”

[1] Zhou, Y., Zheng, D., Mo, Q., Lu, R., Lin, K. Y., & Zheng, W. S. (2025). Decoupled distillation to erase: A general unlearning method for any class-centric tasks. In Proceedings of the Computer Vision and Pattern Recognition Conference (pp. 20350-20359).

**Questions:**

- TA requires computing diag FIM on datasets at $\theta_0$. Therefore, the cost increases linearly with the number of parameters and the size of the data. I think it's hard to use it practically in a large model.

- Recent evidence suggests that MIA scores tend to become less informative as model capacity grows. Experiments on the large model are not in this paper, but what do you think about this?

---

> ### Author Response · Authors · 2025-11-21
>
> Dear Reviewer,
>
> We thank you for the detailed review of our work! You raised some valid concerns in your review, which we have addressed below:
>
> **On including contemporary unlearning methods**
>
> Thank you for pointing this out, this is an ongoing theme amongst reviewers. We are committed to adding additional reference methods to better assess the performance of TA. We appreciate the suggestion of the DELETE [1] paper and recognize that it is relevant related work. However, we do notice a mismatch between the DELETE method specifically and our evaluation protocol.
>
> The DELETE paper considers a more general setting since it does not require access to \\( \\mathcal{D}_r\\) at unlearning time. However, DELETE is only applicable to cases where \\(\\mathcal{D}_f\\) contains an entire class. Hence, this method cannot be applied in the presented sub-class forgetting scenario as well as in the corrupted setting, as described in our paper. The setting considered in DELETE is equivalent to the one in UNSIR [2].
>
> This does not take away from the reviewers point, however, that comparisons with recent work are warranted. To this end, we will add LoTUS [3], to our methods comparison in Table 3.
>
> **"On the assumption of full-access"**
>
> Several contemporary unlearning methods assume a "full access" setting where \\(\\mathcal{D}_r\\) is available at unlearning time. Moreover, the methods which can omit this assumption are often limited to only being able to forget entire classes (UNSIR, DELETE). SSD stands in contrast to this since it pre-computes the diagonal FIM wrt. the training set \\(\\mathcal{D}\\) "offline". In principle, TA can work with a representative subset of \\(\\mathcal{D}_r\\) granted that:
>
> * One pre-computes the FIM for \\(\\mathcal{D}\\) as a preprocessing step, prior to receiving an unlearning request.
> * Since we only sample minibatches from \\(\\mathcal{D}_r\\), as opposed to SCRUB+R, we don’t strictly require all of \\(\\mathcal{D}_r\\).
>
> Although we did not directly experiment with this, it is a simple extension that an organization implementing TA could readily incorporate.
>
> **"In Table 3, please clarify the boldface rule for Time. If SSD (or any other method) is faster but fails the unlearning criteria, consider (a) bolding the fastest among methods that meet success criteria, with those criteria stated in the caption, and (b) visually flagging failed settings (e.g., gray or a footnote). This prevents misinterpretation of boldface as the absolute column minimum."**
>
> We thank the reviewer for pointing this out and have, based on your proposal formulated a concrete bolding criteria for Table 3. This has been inserted in the table caption in the newest revision of the paper.
>
> **"TA requires computing diag FIM on datasets at \\(\\theta_o\\). Therefore, the cost increases linearly with the number of parameters and the size of the data. I think it's hard to use it practically in a large model."**
>
> The cost of computing the diagonal of the FIM is equivalent to performing one training epoch on the training set. Please see Equation 4 in the paper. This is because only first-order derivatives are required when computing the diagonal. As such, its computation can scale to very large models and will in itself never exceed the cost of retraining.
>
> **"Recent evidence suggests that MIA scores tend to become less informative as model capacity grows. Experiments on the large model are not in this paper, but what do you think about this?"**
>
> We recognize that MIA metrics are by no means perfect. Part of the motivation for including MIA results was to stay consistent with prior unlearning literature and give insight into these dynamics in the setting with suboptimal hyperparameters. The relationships between model size and informativeness of MIA scores is interesting. Can you please point to which paper(s) you are referring to? This naturally has to be considered when we are to scale TA to larger model and dataset sizes.
>
> **References:**
>
> [1] Zhou, Y., Zheng, D., Mo, Q., Lu, R., Lin, K. Y., & Zheng, W. S. (2025). Decoupled distillation to erase: A general unlearning method for any class-centric tasks. In Proceedings of the Computer Vision and Pattern Recognition Conference (pp. 20350-20359).
>
> [2] Ayush K Tarun, Vikram S Chundawat, Murari Mandal, Mohan Kankanhalli (2024). Fast Yet Effective Machine Unlearning. IEEE Transactions on Neural Networks and Learning Systems (pp. 13046–13055).
>
> [3] Christoforos N. Spartalis, Theodoros Semertzidis, Efstratios Gavves, Petros Daras (2025). LoTUS: Large-Scale Machine Unlearning with a Taste of Uncertainty. In Proceedings of the Computer Vision and Pattern Recognition Conference (pp. 10046-10055).

---

> > ### Comment · Reviewer_PXQ3 · 2025-11-24
> >
> > Thank you for the author's detailed response.
> >
> > **Contemporary unlearning methods**
> >
> > Of course, TA has a great advantage in that it can handle a variety of unlearning scenarios, unlike existing techniques that deal with some unlearning tasks. However, it seems necessary to compare each task with the state-of-the-art techniques of each task to solve the question of whether TA can produce sufficiently useful performance. I don't think it is a paper with a high contribution only when the performance is higher than the state-of-the-art technique. However, it is better to compare appropriately to ensure that it is useful in all the unlearning tasks presented in this paper. Because, in reality, when a service provider performs an unlearning, it may be advantageous to select the best unlearning technique for each task.
> >
> > Finally, it seems necessary to cite recent unlearning papers appropriately in terms of knowledge distillation-based unlearning. [2, 3, 4]
> >
> >
> > **About MIA,**
> >
> > From Duan et al. [1], they conclude that in the large model, the MIA remains at the level of random guesses at the level of coin tossing. This stems from the nature that large models have fewer training iterations on specific data and are forced to have weak memories of specific data that MIA detects. It is still doubtful whether the MIA is valid as an indicator of the usefulness of TA.
> >
> > + In addition, it is required to properly adjust the position of the components of the paper. For example, the space under Table 1 is too wide.
> >
> > [1] Duan, M., Suri, A., Mireshghallah, N., Min, S., Shi, W., Zettlemoyer, L., ... & Hajishirzi, H. (2024). Do membership inference attacks work on large language models?. arXiv preprint arXiv:2402.07841.
> >
> > [2] Zhou, Y., Zheng, D., Mo, Q., Lu, R., Lin, K. Y., & Zheng, W. S. (2025). Decoupled distillation to erase: A general unlearning method for any class-centric tasks. In Proceedings of the Computer Vision and Pattern Recognition Conference (pp. 20350-20359).
> >
> > [3] Kim, H., Lee, S., & Woo, S. S. (2024, March). Layer attack unlearning: Fast and accurate machine unlearning via layer level attack and knowledge distillation. In Proceedings of the AAAI Conference on Artificial Intelligence (Vol. 38, No. 19, pp. 21241-21248).
> >
> > [4] Ha, S., Park, S., & Yoon, S. W. (2025). Unlearning's Blind Spots: Over-Unlearning and Prototypical Relearning Attack. arXiv preprint arXiv:2506.01318.

---

> > > ### Author Response · Authors · 2025-11-24
> > >
> > > We appreciate your swift response!
> > >
> > >
> > > **On contemporary unlearning methods**
> > >
> > > We agree that it would be ideal to consider the state-of-the-art method within the respective unlearning settings. However, assessing this is non-trivial and would require additional research beyond the scope of the paper. One of the main arguments in our work is that for an unlearning method to practically applicable, it must perform close to retraining on a broad range of hyperparameters. Our claim is not that TA sets a new state-of-the-art on the unlearning tasks. Rather, we argue that hyperparameter sensitivity is a key limiting factor to the practical applicability of unlearning methods and canonical methods, SCRUB+R and SSD, are heavily influenced by the choice of these.
> > >
> > > Concretely:
> > > * SSD is greatly influenced by \\( \\alpha \\), the threshold at which a model parameter is intervened on (see line 194). As  \\( \\alpha \\rightarrow 0 \\), catastrophic unlearning occurs meanwhile as \\( \\alpha \\rightarrow \\infty \\), the SSD rule becomes the identity function.
> > >
> > > * SCRUB+R is highly sensitive to the number of forget and repair epochs which is evident from the results in Figure 1 as well as the newly added Proposition 1.
> > >
> > > Meanwhile, TA remains robust to the choice of forget and repair epochs (evident from corollary 1 and Figure 1). TA performing well across forget set types (Table 3) of course highlights that the method is versatile. But in light of the evaluation protocol used to produce Table 3, the performance of TA shows that it is robust.
> > >
> > > We do agree that comparing to more contemporary methods is a strength which is why we are in the process of adding LoTUS to Table 3. However, we believe that, with this addition, the paper brings sufficient novelty to the field of unlearning.
> > >
> > >
> > > **On extending related work**
> > >
> > > Thank you for pointing us to [3] and [4] (and [2] in your previous response)! We will definitely add [2] to related work as part of the next revision.
> > >
> > > While [3] is based on Knowledge Distillation, we notice that the method considers only intervening on the final model layer, prior to the classification head, rather than the entire model. This leans in the direction of output suppression since dormant information about the forget set will still be contained in the previous layers. Although our paper is concerned with functional equivalence to retraining, rather than representational equivalence, this remains a notable shift from our setting.
> > >
> > > The primary focus of [4] seems to be evaluating unlearning and introduces an over-unlearning metric as well as the prototypical relearning attack. While the Spotter method builds on knowledge distillation, this appears more as a novel framework for addressing these concerns. Introducing this method in related work might therefore be more suitable in the context of unlearning evaluation, a less emphasized part of our work.
> > >
> > >
> > > **On MIA**
> > >
> > > Thank you for highlighting this - it aligns well with the observation that model confidence increases as a function of epochs. However, this a limitation of MIA broadly rather than specifically related to TA.
> > >
> > > Regarding this statement:“It is still doubtful whether the MIA is valid as an indicator of the usefulness of TA.”
> > > Can the reviewer highlight why MIA would be less valid for TA specifically as opposed to e.g. SCRUB+R or SSD?
> > >
> > >
> > > **”In addition, it is required to properly adjust the position of the components of the paper. For example, the space under Table 1 is too wide.”**
> > >
> > > Thank you for pointing to this, we will update this in the next revision when adding results for LoTUS.

---

### Official Review · Reviewer_y2by · 2025-10-30

**Soundness:** 2
**Presentation:** 3
**Contribution:** 2
**Rating:** 2
**Confidence:** 3

**Summary:**

The paper introduces Teacher Ascent (TA), a novel fine-tuning–based machine unlearning method that removes the influence of specific data subsets without full retraining. TA integrates knowledge distillation with Elastic Weight Consolidation (EWC), using a Fisher Information–based regularizer to preserve parameters essential for generalization while bounding the KL-divergence to prevent catastrophic forgetting. Compared to prior methods like SCRUB+R and SSD, TA exhibits strong robustness to hyperparameter variations and avoids instability caused by unbounded divergence terms. Experiments on MNIST, CIFAR-10/100, and Pins Face Recognition show that TA achieves performance comparable to a retrained model while being more computationally efficient.

**Strengths:**

The paper is well-motivated, clearly identifying the limitations of the previous state-of-the-art method SCRUB+R, particularly its instability and sensitivity to hyperparameters, and introducing a simple yet effective solution through the proposed Teacher Ascent (TA) framework.

TA demonstrates strong robustness, stability, and efficiency in unlearning tasks. As shown in Figure 1, TA maintains performance comparable to retraining while avoiding the catastrophic forgetting observed in SCRUB+R. Figure 2 illustrates TA’s low accuracy variance across a wide range of hyperparameters, highlighting its robustness and ease of tuning. Table 3 provides empirical evidence of TA’s significant computational efficiency, achieving much faster running times than SCRUB+R while maintaining similar unlearning quality.

**Weaknesses:**

Novelty: While the paper presents a well-engineered improvement over SCRUB+R, the novelty and insight appear somewhat incremental. TA essentially extends SCRUB+R by adding two additional components: (1) an entropy maximization term (Eq. 5) to enhance forgetting stability, and (2) an EWC-style FIM regularization with dampening factor (proposed by SSD) to preserve critical parameters. Although these modifications yield practical improvements, they mainly combine existing techniques rather than introducing a fundamentally new theoretical perspective or algorithmic paradigm.

Contribution: The paper does not provide deeper analysis or theoretical justification explaining why these components effectively mitigate SCRUB+R’s drawbacks. As a result, the insight and interpretability of the proposed approach remain limited. Incorporating theoretical analysis could strengthen the contribution and clarify the underlying mechanism behind TA.

Experiment: The paper compares TA against SCRUB+R (2023) and SSD (2024) only, which does not convincingly demonstrate the effectiveness of TA. Given that the venue will be held in 2026, the author needs to compare it against broader and more recent unlearning baselines.

**Questions:**

1. Can the authors explain why the proposed terms can improve the robustness of hyperparameter selection?
2. Why does TA compute faster than SCRUB+R, given that they are similar? Is that because the k was set to 1 in TA?
3. Since SCRUB+R is sensitive to hyperparameters, what is the strategy of its hyperparameter selection? Since they are searched by TPE, does that affect and lower the performance of it?

---

> ### Author Response · Authors · 2025-11-21
>
> Dear Reviewer,
>
> We thank you for the thorough review of our work. As part of your feedback, you raised several concerns which we have addressed these in the following. We believe that this revision has greatly strengthened the quality of our work.
>
> **"Incremental novelty: ...(2) an EWC-style FIM regularization with dampening factor (proposed by SSD) to preserve critical parameters. Although these modifications yield practical improvements, they mainly combine existing techniques rather than introducing a fundamentally new theoretical perspective or algorithmic paradigm."**
>
> In the original submission, we accidentally flipped the order of  \\(\\mathcal{D}_r \\) and \\(\\mathcal{D}_f \\) in the SSD decision rule (line 193). The correct formula is:
> \\[
> \\theta_j = \\begin{cases}
>         \\beta \\cdot \theta_j & f^{(\\mathcal{D}_f)}\_{j} > \\alpha f^{(\\mathcal{D}\_r)}\_{j} \\\\
>         \\theta_j & \text{otherwise}
>     \\end{cases}
> \\]
> which has been corrected in the newly revised version. With the information you had available at the time, you are correct that the EWC regularization term effectively becomes the same as the SSD rule. However, in actuality the EWC term has the opposite role of SSD. Where SSD identifies parameters important for the forget set, the FIM ratio identifies parameters important for retaining knowledge. To the best of our knowledge, this has not been applied in prior work.
>
> **"Contribution: The paper does not provide deeper analysis or theoretical justification explaining why these components effectively mitigate SCRUB+R’s drawbacks."**
>
> The main drawback in SCRUB+R, which TA addresses, is its sensitivity to the number of forget and retain epochs. We have taken this critique to heart, and have formulated a detailed proof showing that:
>
> * If SCRUB+R is run too long, the norm of the student parameters will diverge.
>
> * Bounding student probabilities to \\( \\epsilon \\) e.g., \\(q\_i = \\max(\\epsilon, q\_i)\\), effectively stops gradients for the logits corresponding to classes \\(c\\) where \\(q\_c \\leq \\epsilon\\). A direct consequence of this is that the norm of student parameters remains finite in the limit.
>
> These findings have been integrated into the main paper (top of section 3). Detailed proof of this is given in Appendix A of the new revision.
>
> **"Experiment: The paper compares TA against SCRUB+R (2023) and SSD (2024) only, which does not convincingly demonstrate the effectiveness of TA. Given that the venue will be held in 2026, the author needs to compare it
> against broader and more recent unlearning baselines."**
>
> This point resonates with us, and we are committed to comparing TA with newer unlearning literature. Specifically, we will add LoTUS [1] as another baseline. We are currently in the process of hyperparameter tuning LoTUS and will report results as soon as possible.
>
> **"Can the authors explain why the proposed terms can improve the robustness of hyperparameter selection?"**
>
> This observation is partly empirical and stems from the design of the evaluation suite. Since hyperparameter search is done on different forget sets than what we evaluate on, consistently mimicking a retrained model in this scenario implies robustness towards the selected hyperparameters.
>
> Beyond this, the newly added proof that SCRUB+R makes model parameters diverge in the limit supports this claim. Through this analysis, we have a theoretical explanation as to why TA is less sensitive than SCRUB+R to the choices of the number of forget and retain rounds. The results in Figure 1 highlight this empirically.
>
> **"Why does TA compute faster than SCRUB+R, given that they are similar? Is that because the k was set to 1 in TA?"**
>
> Yes, TA scales with \\(k \\cdot |\\mathcal{D}\_f|\\), and we use \\(k=1\\) in all experiments. SCRUB+R by design iterates over the whole retain dataset per round. Because \\(|\\mathcal{D}\_f| \\ll |\\mathcal{D}\_r| \\), we get a notable difference in runtime.
>
> **"Since SCRUB+R is sensitive to hyperparameters, what is the strategy of its hyperparameter selection? Since they are searched by TPE, does that affect and lower the performance of it?"**
>
> The strategy for finding hyperparameters for SCRUB+R is identical to that of TA and SSD e.g., we search for parameters that best match a retrained model's forget and retain accuracy on a forget set semantically related to the downstream task. We see no indication, theoretically or empirically, that using TPE for determining appropriate parameters for SCRUB+R could yield artificially low results.
>
> **references:**
>
> [1] Christoforos N. Spartalis, Theodoros Semertzidis, Efstratios Gavves, Petros Daras (2025). LoTUS: Large-Scale Machine Unlearning with a Taste of Uncertainty. Proceedings of the IEEE/CVF Conference on Computer Vision and Pattern Recognition (CVPR), 2025, pp. 10046-10055.

---

### Official Review · Reviewer_3ohU · 2025-10-30

**Soundness:** 2
**Presentation:** 3
**Contribution:** 3
**Rating:** 6
**Confidence:** 2

**Summary:**

This paper proposes Teacher Ascent (TA), a robust and efficient machine unlearning method. Its main strength is a novel design that addresses the hyperparameter sensitivity of prior work, demonstrated through a more realistic evaluation protocol. However, the experimental validation is limited to small-scale datasets, which raises concerns about its real-world applicability.

**Strengths:**

Novel and Robust Method: TA's design, featuring bounded KL-divergence and a discriminative EWC regularizer, is a novel and effective solution to the instability problems of prior SOTA methods.
Practical Evaluation Protocol: The proposed protocol of using a proxy task for hyperparameter tuning is innovative and better reflects real-world constraints.
Strong Performance: On the tested benchmarks, TA achieves performance comparable to the gold-standard retraining but is significantly more efficient and robust.

**Weaknesses:**

Formatting Errors: There are multiple formatting issues: missing commas after "e.g.", missing punctuation for equations, unnumbered equations, and the caption for Table 2 is incorrectly placed below the table.
Limited Experimental Scale: All experiments are conducted on small-scale datasets (CIFAR-10/100). This is a major concern, as it's unclear if the method's effectiveness and efficiency gains will generalize to the large-scale models and larger forget sets common in real-world applications. Experiments on a larger scale are needed.

**Questions:**

My primary concern, which will heavily influence my final recommendation, is the limited scale of the experimental validation. The paper makes strong claims about real-world applicability and robustness, yet the empirical evidence is exclusively based on small-scale datasets like CIF-AR-10/100, where forget sets are only a few hundred images. This raises critical questions about the method's generalizability.
To address this limitation, I would like the authors to respond to the following points during the rebuttal phase. A convincing response here could significantly change my opinion of the paper.

---

> ### Author Response · Authors · 2025-11-21
>
> Dear Reviewer,
>
> Thank you for taking the time to go over our work! You highlighted some concerns in your response which we have addressed below:
>
> **Formatting Errors: "There are multiple formatting issues: missing commas after "e.g.", missing punctuation for equations, unnumbered equations, and the caption for Table 2 is incorrectly placed below the table."**
>
> Thank you for pointing this out! We have proof-read the manuscript and corrected several typos and formatting mistakes, including the ones which you highlight.
>
> **"Limited Experimental Scale: All experiments
> are conducted on small-scale datasets (CIFAR-10/100). This is a major concern, as it's unclear if the method's effectiveness and efficiency gains will generalize to the large-scale models and larger forget sets common in real-world applications. Experiments on a larger scale are needed."**
>
> Thank you for highlighting this, we agree that scaling to larger datasets and model sizes would strengthen the claims surrounding real-world applicability. However, we do not see it being feasible that we, within the time-frame of the review period, can experiment with increased model/data sizes. This is due to both time and compute constraints.
>
> That being said, we hope that the following additional contributions can satisfy the reviewer on this front:
>
> * We will add LoTUS [1] to the comparison in Table 3. This will give insight into how TA compares to more contemporary research and further map out the literature on unlearning with suboptimal hyperparameters.
>
> * We have, based on the response of reviewer y2by, formulated a proof that SCRUB+R will cause model parameters to diverge in the limit. Further, we show that bounding probabilities used in KL-divergence maximization, as done in TA, stops gradients for classes in the *unlearning limit*. This directly addresses the divergence of SCRUB+R and also provides a theoretical perspective on the results reported in Figure 1. This result also gives perspectives on whether TA will remain stable in large-scale contexts. We have added this as part of section 3 in the revised paper and provide extensive proof in Appendix A.
>
> Another aspect surrounding the generalizability of efficiency gains to large-scale settings is that the runtime of TA scales with the size of the forget set, \\( \\mathcal{D}_f \\). This follows from the fact that minibatches from retain data, \\( \\mathcal{D}_r \\), are sampled during unlearning (line 13, Algorithm 1). This stands in contrast to SCRUB+R which iterates over the entire training set per "unlearning epoch". Since \\(|\\mathcal{D}\_f| \\ll | \\mathcal{D}\_r| \\), the scaling properties of TA are much more desirable.
>
> **references**
>
> [1] Christoforos N. Spartalis, Theodoros Semertzidis, Efstratios Gavves, Petros Daras (2025). LoTUS: Large-Scale Machine Unlearning with a Taste of Uncertainty. Proceedings of the IEEE/CVF Conference on Computer Vision and Pattern Recognition (CVPR), 2025, pp. 10046-10055.

---

### Official Review · Reviewer_YRo7 · 2025-10-31

**Soundness:** 1
**Presentation:** 2
**Contribution:** 2
**Rating:** 4
**Confidence:** 3

**Summary:**

The paper proposes Teacher Ascent (TA), a machine unlearning method designed to remove specific knowledge from a trained model without retraining it from scratch. This work improves upon SCRUB+R, which applies knowledge distillation to align the student’s output distribution with the original model on the retain dataset while encouraging a uniform output on the forget dataset. In addition, it introduces a continual learning inspired regularization based on Elastic Weight Consolidation (EWC) to selectively preserve parameters important to the retain dataset, preventing catastrophic forgetting.

**Strengths:**

The methodology section is clearly written and well-structured. The logical flow of the motivation and related work is coherent and easy to follow, making the paper accessible even to readers who are new to the field.

**Weaknesses:**

However, the experimental section lacks sufficient clarity. For example, the authors do not adequately explain each line in Figure 1, making it difficult for readers to interpret the results. Moreover, the results reported in Table 3 are somewhat confusing. It is unclear whether higher or lower values are desirable for each metric, and the rationale for highlighting certain results in bold as the best outcomes is not well explained.

**Questions:**

The results presented in Table 3 appear unusual. For instance, the Forget Accuracy is generally expected to decrease; however, in the corrupted setting of CIFAR-10, the Teacher Ascent (TA) method achieves a Forget Accuracy of 1.00, higher than SCRUB+R and SSD, and it is labeled as the best result. Could the authors clarify this outcome and explain how such a result aligns with the expected behavior of unlearning methods?
In addition, in the whole-class setting of CIFAR-10, the Retain Accuracy values for all methods are reported as 1.00, yet only TA and SCRUB+R are highlighted in bold. The authors are encouraged to clarify the criteria used for emphasizing these results.

---

> ### Author Response · Authors · 2025-11-21
> **Response to reviewer YRo7**
>
> Dear Reviewer,
>
> We thank you kindly for taking the time to review our work! You raised some valid concerns and highlighted sources of confusion, which we addressed in the following:
>
> **"Lack of explanation of lines in figure 1":**
> You are correct, there was supposed to be a legend for figure 1 explaining each line. This was accidentally removed. In the newly uploaded version a legend has been inserted.
>
>
> **"results reported in Table 3 are confusing. It is unclear whether higher or lower values are desirable for each metric, and the rationale for highlighting certain results in bold as the best outcomes is not well explained"**
>
>
> In general, when evaluating unlearning, the ideal model would be functionally equivalent to a retrained model. As such, the results in Table 3 for the unlearning methods per forget set should be compared to the associated retrained model. Note, that it is not sufficient that the unlearned model matches retraining on one dataset.
>
> Consider forgetting a single observation where the original model correctly classifies the sample and the retrained model misclassifies that observation. Let the unlearning method be the identity function. Then we would have that it matches retraining nearly perfectly on retain/validation data, yet the forget accuracy is maximally wrong.
>
> **"...in the corrupted setting of CIFAR-10, TA achieves a Forget Accuracy of 1.00, higher than SCRUB+R and SSD, and it is labeled as the best result. Could the authors clarify this outcome and explain how such a result aligns with the expected behavior of unlearning methods?"**
>
> We recognize that it is not immediately clear from the manuscript how results in the corrupted setting should be interpreted. To this end, we provided further explanation of the corrupted case in the caption of Table 3. To avoid any further confusion, we address how the results should be interpreted.
>
> In the corrupted setting, the accuracy is reported wrt. the true label for corrupted samples. That is, for forget samples belonging exclusively to class 1 that were mislabeled as class 2, the accuracy on the forget set is:
>
> \\[
> acc\_{\\mathcal{D}\_f} = \\frac{1}{|\\mathcal{D}\_f|} \\sum\_{(\\boldsymbol{x}\_i, y\_i) \\in \\mathcal{D}\_f} \\mathbb{1}(\\mathcal{M}(\\boldsymbol{x}\_i) = 1)
> \\]
>
> where \\(\\mathbb{1}(\\cdot) \\) is the indicator function. For corrupted data, the retrained model achieved \\(acc\_{\\mathcal{D}\_f} = 0.98 \\pm 0.01\\) whereas TA achieves \\(acc_{\mathcal{D}_f} = 1.00 \pm 0.01\\). On the contrary, the mean accuracy of SCRUB+R is \\(8\\%\\)-points away from the retrained model. On the retain and validation data, TA is also closer the retrained model than SCRUB+R. Notice here that the original model's performance on the forget set is \\(0.16 \\pm 0.05 \\) meaning that it overfitted to the mislabeled class for the forget samples. Further details on how the experiment with corrupted data was carried out are in Appendix A.1.2 of the original submission.
>
>
> **"In the whole-class setting of CIFAR-10, the Retain Accuracy
> values for all methods are reported as 1.00, yet only TA and SCRUB+R are highlighted in bold. The authors are encouraged to clarify the criteria used for emphasizing these results."**
>
> We agree that the criteria for when certain results are highlighted in bold should be stated explicitly. Bolding results is meant as a help to the reader for identifying which method(s) were the closest to retraining granted that unlearning was adequate. However, we recognize that this is a source of confusion for the reader without stating explicit criteria. To this end, we have revised the bolding of results in Table 3 and explicitly stated the criteria in the caption.
>
> The reason why SSD's retain accuracy is not highlighted in the whole-class setting on CIFAR-10 is because it did not unlearn *adequately*, achieving a mean forget accuracy of \\(91 \\% \\) whereas retraining had \\(0\\% \\) forget accuracy. This is now directly justified under the new highlighting criteria.
>
> We hope that this clears any confusion about how the reported results should be interpreted. Please reach out if any aspects of our work remain unclear, and we will do our best to get back promptly!

---

> ### Comment · Reviewer_YRo7 · 2025-11-25
>
> The authors have addressed my initial concerns. However, I agree with the points raised by other reviewers that the experiments included in the main paper are too simple to demonstrate superior performance compared to current baselines. Consequently, I will raise my score if the large-scale experiment yields positive results.

---

### Comment · Area_Chair_tnSe · 2025-11-22

Dear reviewers,
Please check the authors’ responses. As there are differing opinions about the paper, it would be appreciated if you could evaluate—based on all comments—whether the authors have adequately addressed the main concerns.
br,

---

### Author Response · Authors · 2025-12-01
**Executive summary of rebuttal period**

Dear Area Chair,

Due to the unfortunate circumstances surrounding reviews, we provide you a concise summary of what has happened in the rebuttal period. We hope this will make your job easier while reviewing our work.

\\(   \\ \\)

## Main points across reviewers and actions taken

\\(   \\ \\)

**Issue: The paper needs more extensive experiments**
* Here reviewer PXQ3 proposed to benchmark against DELETE [1]. After reading the paper, we realized the method is applicable when forgetting entire classes, making it unsuitable for the sub-class forgetting and corrupted setting (Tables 2 and 3).

**Actions taken:**

* We have added LoTUS [2] as an additional benchmark method in Table 3 to address the concern on a lack of comparison with contemporary unlearning methods. The overall interpretation of the results in Table 3 remain unchanged; TA emerges as the most viable unlearning method.

* We have provided formal proof that running SCRUB+R for *too long* will cause the model to diverge (proposition 1). Also, we prove that bounding probabilities, as done in Teacher Ascent, directly addresses this shortcoming of SCRUB+R (corollary 1). These have been integrated into the main text (top of section 3) while the entire proof is provided in Appendix A. These findings on the convergence of the methods are also highlighted empirically in Figure 1. This significantly strengthens the claims surrounding hyperparameter robustness of TA.

* Initially we described SCRUB+R as the state-of-the-art unlearning method. While several prior papers have done this, we recognize that this description can be misleading due to the various unlearning settings and inherent stochasticity when comparing unlearning methods. To this end, we have rephrased the wording throughout the main paper. We also have emphasized throughout the review period that this paper is not a benchmark study. Rather it highlights a shortcoming of canonical unlearning methods, namely sensitivity to hyperparameter selection. TA is a potential solution to this problem in light of the novel evaluation protocol. While comparison with more unlearning methods is desirable, and something we address with the addition of LoTUS, it is not a main contribution of the paper.

\\(   \\ \\)

**Issue: The paper lacks novelty and theoretical insights into why the proposed method addresses hyperparameter robustness**

**Actions taken:**

* There was a typo in our initial submission in the SSD update rule formula (line 194) which would give the claim about a lack of novelty more merit. We promptly fixed this and informed the reviewer.

* The previously mentioned proof directly addresses the lack of theoretical insight.

\\(   \\ \\)

**Issue: Confusion about differences in runtime for TA and SCRUB+R**

**Actions taken:**

* We described the difference in runtime explicitly in the answers to reviewers and those who got back said that their concerns were addressed. Concretely, the reason for the difference in runtime is that TA scales with the size of the forget set, \\( \\mathcal{D}\_f \\), whereas SCRUB+R by design iterates over the retain set \\( \\mathcal{D}\_r \\) per unlearning epoch. Since \\( ||\\mathcal{D}\_f|| \\ll \\mathcal{D}\_r \\) in all practical unlearning scenarios, TA's scaling properties are far more desirable.

\\(   \\ \\)

**Issue: Confusion about how to interpret the provided results.**

**Actions taken:**

* We explained how results should be interpreted extensively and the reviewer said that they had no further concerns surrounding this.

\\(   \\ \\)

**Issue: Minor typos pointed out, formatting errors, and confusion about the criteria for when a result was highlighed in bold in Table 3**

**Actions taken:**

* We have proofread the paper several times and addressed all the typos that reviewers highlighted as well as additional ones.

* We have added explicit criteria for when a result is in bold in Table 3 as well as explained the initial justification. No reviewers mentioned further concerns on this.

\\(   \\ \\)

We thank the area chair for your time and will answer any further questions that you might have about our work as swiftly as possible!

\\(   \\ \\)

**References**

[1] Zhou, Y., Zheng, D., Mo, Q., Lu, R., Lin, K. Y., & Zheng, W. S. (2025). Decoupled distillation to erase: A general unlearning method for any class-centric tasks. In Proceedings of the Computer Vision and Pattern Recognition Conference (pp. 20350-20359).

[2] Christoforos N. Spartalis, Theodoros Semertzidis, Efstratios Gavves, Petros Daras (2025). LoTUS: Large-Scale Machine Unlearning with a Taste of Uncertainty. In Proceedings of the Computer Vision and Pattern Recognition Conference (pp. 10046-10055).

---

### Meta-Review · Area_Chair_8kBA · 2026-01-06

**Summary:**

The most consistent issue across reviewers is the limited experimental scale. All empirical results are on small benchmarks (MNIST, CIFAR-10/100), with small forget sets. Multiple reviewers explicitly stated that this makes it difficult to judge whether Teacher Ascent would work for the larger models and larger, more complex unlearning requests that motivate the paper's claims about practicality and robustness. The rebuttal acknowledged this limitation but did not provide new evidence, leaving a gap between the claims and the validation.

A second major concern is novelty and conceptual contribution. Several reviewers felt that Teacher Ascent is a well-engineered refinement of existing ideas (knowledge distillation–based unlearning, bounded divergence, and EWC/Fisher regularization), rather than a fundamentally new approach. While the added theory helps explain why the method is more stable than SCRUB+R, it does not fully resolve the impression that the contribution is incremental.

Reviewers also raised issues with baseline coverage and evaluation completeness. Comparisons focus mainly on SCRUB+R and SSD. Some reviewers argued that, for each unlearning scenario considered, the paper should be compared against the strongest relevant methods for that setting, rather than relying on a narrow baseline set.

Finally, there are practical realism concerns. The method assumes access to retain data (or at least to Fisher information computed from it) at unlearning time, and relies on MIA as a key evaluation metric, even though several reviewers questioned MIA's reliability for larger models. The rebuttal addressed these points at a high level but did not convincingly demonstrate that they would not become serious issues in realistic deployments.

Taken together, while the paper is clearly written, the combination of limited empirical scope, incremental novelty, and unresolved practical questions are not enough to support acceptance.

**Reviewer Concerns:**

For Reviewer YRo7, the rebuttal mostly addressed what the reviewer actually asked. The authors clarified how to interpret Table 3 (match retraining rather than "forget accuracy must always go down"), explained the corrupted-label setting, and made the bolding criteria explicit. The reviewer even acknowledged that these initial issues were resolved. What's still outstanding from this thread is not YRo7's original question, but their follow-up alignment with others: the experiments in the main paper still feel too small/simple to justify strong claims.

For Reviewer 3ohU, the rebuttal handled the minor formatting issues, but the main concern (limited scale: only CIFAR/MNIST-style settings and small forget sets) is still largely outstanding. The authors basically said they couldn't run large-scale experiments within the review timeline. They tried to compensate with arguments about stability, a proof sketch, and scaling intuition (runtime depends on |D_f| more than |D_r|), but they did not provide the empirical evidence the reviewer asked for. So the central question (does it work on larger models/datasets or more realistic forget requests) remains unanswered.

For Reviewer y2by, the rebuttal made real progress on a few points: they corrected an apparent mistake in how they described the SSD-style term, and they added theory aimed at explaining why bounding the KL term improves stability and why SCRUB+R can diverge. They also gave a clear reason TA can be faster than SCRUB+R (k=1 and scaling with forget set rather than iterating over retain data). That said, the bigger issues are only partially addressed: the novelty still looks incremental (a careful combination of KD + bounded divergence + EWC/FIM ideas), and the baseline coverage is still thin.

For Reviewer PXQ3, the rebuttal responded but didn't really close the loop. They acknowledged the lack of contemporary baselines and argued why DELETE doesn't match all their scenarios, but they didn't add strong task-appropriate SOTA comparisons beyond promising LoTUS. The "full-access" concern (needs retain data / relies on FIM and sampling from D_r) is still outstanding because they mainly argued it could be adapted rather than showing an ablation where D_r access is reduced. The scalability concern about computing FIM for large models was answered with "it's like one epoch," but again without any large-scale demonstration. On MIA, the rebuttal mostly deflects by saying MIA has known limitations in general. The reviewer's follow-up (citing evidence that MIA can be near-random for large models) leaves the evaluation story feeling unresolved.

**Reviewer Scores:**

Reviewer YRo7's original concerns were mostly about clarity and interpretation, and those were genuinely addressed. They explicitly acknowledged that their initial issues were resolved. However, they also aligned themselves with other reviewers on the lack of convincing large-scale experiments. If they had fully participated, I think their score would likely be the same, because their remaining concern depended on evidence that never materialized.

Reviewer 3ohU made it very clear that experimental scale was the deciding factor. The rebuttal did not deliver new large-scale experiments, only explanations for why they were infeasible and some theoretical arguments. That likely would not change their position. At best, their score might remain at 6.

The rebuttal meaningfully addressed some technical points raised by Reviewer y2by: correcting the SSD-related formula, adding a theoretical argument for stability, and clarifying runtime differences. That said, the reviewer's core concerns about incremental novelty and limited baseline coverage remain largely intact. I could see this reviewer softening slightly (e.g., from 2 to 4), but not moving into acceptance territory. The work would still look like a careful refinement of existing ideas rather than a strong conceptual advance.

Reviewer PXQ3 raised deep concerns about baseline completeness, full-access assumptions, scalability of FIM, and the validity of MIA, especially for larger models. The rebuttal responded, but mostly with arguments rather than evidence. Even with full participation, I do not see this reviewer raising their score meaningfully.

---

### Decision · Program_Chairs · 2026-01-26

Reject